# Retrieval of ice nucleating particle concentrations from lidar observations and comparison with UAV in-situ measurements

Eleni Marinou[1,2,3], Matthias Tesche[4,5], Athanasios Nenes[6,7], Albert Ansmann[8], Jann Schrod[9], Dimitra Mamali[10], Alexandra Tsekeri[1], Michael Pikridas[11], Holger Baars[8], Ronny Engelmann[8], Kalliopi - Artemis Voudouri[2], Stavros Solomos[1], Jean Sciare[11], Silke Groß[3], Florian Ewald[3], and Vassilis Amiridis[1]

[1]IAASARS, National Observatory of Athens, Athens, 15236, Greece
[2]Department of Physics, Aristotle University of Thessaloniki, Thessaloniki, 54124, Greece
[3]Institute of Atmospheric Physics, German Aerospace Center (DLR), Oberpfaffenhofen, 82234, Germany
[4]University of Hertfordshire, College Lane, AL10 9AB Hatfield, United Kingdom
[5]Leipzig Institute for Meteorology, Leipzig University, Leipzig, Germany
[6]Laboratory of Atmospheric processes and Their Impact (LAPI), School of Architecture, Civil and Environmental Engineering, École Polytechnique Fédérale de Lausanne, CH-1015, Switzerland
[7]Institute of Chemical Engineering Sciences, Foundation for Research and Technology, Hellas, Patras, 26504, Greece
[8]Leibniz Institute for Tropospheric Research, 04318 Leipzig, Germany
[9]Institute for Atmospheric and Environmental Sciences, Goethe University Frankfurt, 60438, Frankfurt am Main, Germany
[10]Department of Geoscience and Remote Sensing, Delft University of Technology, Delft, The Netherlands
[11]The Cyprus Institute, Energy, Environment and Water Research Centre, Nicosia, Cyprus

*Correspondence to:* Eleni Marinou (elmarinou@noa.gr)

**Abstract.**

Aerosols that are efficient ice nucleating particles (INPs) are crucial for the formation of cloud ice via heterogeneous nucleation in the atmosphere. The distribution of INPs on a large spatial scale and as a function of height determines their impact on clouds and climate. However, in-situ measurements of INPs provide sparse coverage over space and time. A promising approach to address this gap is to retrieve INP concentration profiles by combining particle concentration profiles derived by lidar measurements with INP efficiency parameterizations for different freezing mechanisms (immersion freezing, deposition nucleation). Here, we assess the feasibility of this new method for both ground-based and space-borne lidar measurements, using in-situ observations collected with Unmanned Aerial Vehicles (UAVs) and subsequently analyzed with the FRIDGE (FRankfurt Ice nucleation Deposition freezinG Experiment) INP counter from an experimental campaign at Cyprus in April 2016. Analyzing five case studies we calculated the cloud relevant particle number concentrations using lidar measurements ($n_{250,\mathrm{dry}}$ with an uncertainty of 20 to 40% and $S_{\mathrm{dry}}$ with an uncertainty of 30 to 50%) and we assessed the suitability of the different INP parameterizations with respect to the temperature range and the type of particles considered. Specifically, our analysis suggests that our calculations using the parameterization of Ullrich et al. (2017) (applicable for the temperature range $-50\,^{\circ}\mathrm{C}$ to $-33\,^{\circ}\mathrm{C}$) agrees within one order of magnitude with the in-situ observations of $n_{\mathrm{INP}}$ thus, the parameterization of Ullrich et al. (2017) can efficiently address the deposition nucleation pathway in dust-dominated environments. Additionally, our calculations using the combination of the parameterizations of DeMott et al. (2015) and DeMott et al. (2010) (applicable

for the temperature range $-35\,^{\circ}\mathrm{C}$ to $-9\,^{\circ}\mathrm{C}$) agrees within two orders of magnitude with the in-situ observations of INP concentrations ($n_{\mathrm{INP}}$) and can thus efficiently address the immersion/condensation pathway of dust and continental/anthropogenic particles. The same conclusion is derived from the compilation of the parameterizations of DeMott et al. (2015) for dust and Ullrich et al. (2017) for soot.

Furthermore, we applied this methodology to estimate the INP concentration profiles before and after a cloud formation, indicating the seeding role of the particles and their subsequent impact on cloud formation and characteristics. More synergistic datasets are expected to become available in the future from EARLINET (European Aerosol Research Lidar NETwork) and in the frame of the European ACTRIS-RI (Aerosols, Clouds, and Trace gases Research Infrastructure).

    Our analysis shows that the developed techniques, when applied on CALIPSO (Cloud-Aerosol Lidar and Infrared Pathfinder

Satellite Observations) space-born lidar observations, are in very good agreement with the in-situ measurements. This study gives us confidence for the production of global 3D products of cloud relevant particle number concentrations ($n_{250,\mathrm{dry}}$, $S_{\mathrm{dry}}$ and $n_{\mathrm{INP}}$) using the CALIPSO 13-yrs dataset. This could provide valuable insight into global height-resolved distribution of INP concentrations related to mineral dust, and possibly other aerosol types.

## 1   Introduction

The interaction of aerosol particles with clouds, and the related climatic effects have been in the focus of atmospheric research for several decades. Aerosols can act as cloud condensation nuclei (CCN) in liquid water clouds and as ice nucleating particles (INPs) in mixed-phase and ice clouds. Changes in their concentration affect cloud extent, lifetime, particle size and radiative properties (Lohmann and Feichter, 2005; Tao et al., 2012; Altaratz et al., 2014; Rosenfeld et al., 2014). As important these interactions are, they are the source of the highest uncertainty in assessing the anthropogenic climate change (IPCC Fifth

Assessment Report, Seinfeld et al. 2016).

    All clouds producing ice require, for temperatures above $\sim -35\,^{\circ}\mathrm{C}$, the presence of INPs. Compared to CCN, INPs are rare (about one particle in a million act as INP; Nenes et al. (2014)), and become increasingly sparse with increasing temperature (Pruppacher and Klett, 1997; Kanji et al., 2017). Aerosol species which are identified in the past as potentially important INPs are mineral dust, biological species (pollen, bacteria, fungal spores and plankton), carbonaceous combustion products, soot,

volcanic ash and sea spray (Murray et al., 2012; DeMott et al., 2015b). From these aerosol types, mineral dust and soot are efficient INPs at temperatures below $-15\,^{\circ}\mathrm{C}$ to $-20\,^{\circ}\mathrm{C}$ (dust) and $-40\,^{\circ}\mathrm{C}$ (soot) and they have been studied extensively for their INP properties in field experiments and laboratory studies (Twohy et al., 2009, 2017; Kamphus et al., 2010; Hoose and Möhler, 2012; Murray et al., 2012; Sullivan et al., 2016; Ullrich et al., 2017). Biological particles are one of the most active INP species, however their abundance is likely low on a global scale, particularly when compared to other aerosol types such as mineral

dust (Morris et al., 2014). It has been suggested that soil and clay particles may act as carriers of biological nanoscale INPs (e.g. proteins), which could potentially contribute to a globally/locally source of INP (Schnell and Vali, 1976; O'Sullivan et al., 2014, 2015, 2016). Finally, marine aerosols (with possible influence of a biological microlayer close to the surface) are also

important INPs in areas where the influence of mineral dust is less pronounced (e.g. Southern Ocean; Wilson et al. 2015; Vergara-Temprado et al. 2017).

There is a variety of pathways for heterogeneous ice nucleation: contact freezing, immersion freezing, condensation freezing and deposition nucleation (Vali et al., 2015). Individual ice nucleation pathways dominate at characteristic temperatures and su-
5 persaturation ranges. Observational studies have shown that immersion freezing dominates at temperatures higher than $-30\,°C$, while deposition nucleation dominates below $-35\,°C$ (Ansmann et al., 2008, 2009; Westbrook et al., 2011; de Boer et al., 2011). The factors that regulate the efficiency of heterogeneous ice nucleation are qualitatively understood, but no general theory of heterogeneous ice nucleation exists yet. It has been shown that in regions not influenced by sea salt aerosol, INP concentrations are strongly correlated with the number of aerosol particles with dry radius greater than $250\,nm$ ($n_{250,dry}$) which
form the reservoir of favorable INPs (DeMott et al., 2010, 2015). However, we have limited knowledge on how the ice nuclei activity of these particles together with their spatial and vertical distributions depend on cloud nucleation conditions (i.e. temperature ($T$) and supersaturation over water ($ss_w$) and ice ($ss_i$)). Furthermore, field measurements of INP concentrations are very localized in space and time, whilst there are large regions without any data at all (Murray et al., 2012). The lack of data inhibits our quantitative understanding of aerosol-cloud interactions and requires new strategies for obtaining datasets
(Seinfeld et al., 2016; Bühl et al., 2016).

Active remote sensing with aerosol lidar and cloud radar provides valuable data for studying aerosol-cloud interaction since it enables observations with high vertical and temporal resolution over long time periods (Ansmann et al., 2005; Illingworth et al., 2007; Seifert et al., 2010; de Boer et al., 2011; Kanitz et al., 2011; Bühl et al., 2016). Lidar measurements can provide profiles of $n_{250,dry}$ (the number of aerosol particles with dry radius greater than $250\,nm$) and $S_{dry}$ (the aerosol particles dry surface
area concentration) related to mineral dust, continental pollution and marine aerosol, as described in Mamouri and Ansmann (2015, 2016). Their methodology uses lidar-derived optical parameters (i.e. the particle backscatter coefficient, lidar ratio and particle depolarization ratio) to separate the contribution of mineral dust in the lidar profiles (Tesche et al., 2009) and subsequently applies sun-photometer based parameterizations to transform the optical property profiles into profiles of aerosol mass, number, and surface-area concentration (Ansmann et al., 2012; Mamouri and Ansmann, 2015, 2016). The latter can then
be used as input to INP parameterizations that have been obtained from laboratory and field measurements (e.g. DeMott et al. 2010; Niemand et al. 2012; DeMott et al. 2015; Steinke et al. 2015; Ullrich et al. 2017) to derive profiles of INP concentrations ($n_{INP}$).

The INP retrieval calculated from the lidar measurements provides a promising insight into atmospheric INP concentrations. To date, there has been no other evaluation of the lidar-derived profiles of $n_{250,dry}$, $S_{dry}$ and $n_{INP}$ by means of independent in-situ
observations apart from one dust case in Schrod et al. (2017). The study presented here compares $n_{250,dry}$ and $n_{INP}$ as inferred from space-borne and ground-based lidar observations to findings from airborne in-situ measurements using data from the joint experiment "INUIT-BACCHUS-ACTRIS" (Ice Nuclei Research Unit - Impact of Biogenic versus Anthropogenic emissions on Clouds and Climate: towards a Holistic UnderStanding - Aerosols, Clouds, and Trace gases Research Infrastructure) held on April 2016 in Cyprus (Schrod et al., 2017; Mamali et al., 2018). The paper starts with a review of the different INP parame-
terizations for mineral dust, soot and continental aerosols in Section 2. Section 3 describes the instruments used in this study

and the methodology to retrieve INP concentrations from lidar measurements. The results of the intercomparison between the lidar-derived and Unmanned Aerial Vehicle (UAV) measured $n_{250,\text{dry}}$ and $n_{\text{INP}}$ profiles are presented and discussed in Section 4 before the paper closes with a summary in Section 5.

## 2  INP parameterizations

A variety of parameterizations has been proposed to obtain $n_{\text{INP}}$ from aerosol concentration measurements. In particular, a global aerosol type-independent $n_{\text{INP}}$ parameterization was introduced by DeMott et al. (2010), dust-specific $n_{\text{INP}}$ parameterizations were introduced by Niemand et al. (2012); DeMott et al. (2015); Steinke et al. (2015); Ullrich et al. (2017) and soot-specific $n_{\text{INP}}$ parameterizations were proposed by Murray et al. (2012) and Ullrich et al. (2017). The aforementioned parameterizations address immersion freezing at or above water saturation and deposition nucleation for ice saturation ratios

ranging from unity up to the homogeneous freezing threshold and water saturation. Table 1 provides an overview of the temperature ranges and the freezing mechanisms for which these parameterizations are applicable.

Regarding immersion freezing, the aerosols that are activated to droplets can contribute to ice formation. In turn, the ability of a particle to be activated as a cloud droplet mainly depends on the cloud supersaturation, its diameter, the water adsorption characteristics and the composition of soluble coatings (Levin et al., 2005; Kumar et al., 2011a, b; Garimella et al., 2014;

Begue et al., 2015). Kumar et al. (2011b) showed that all dry-generated dust samples with radius $> 50\,\text{nm}$ are activated to CCN at water supersaturation ($ss_{\text{w}}$) of 0.5% while the activation radius increases to $> 250\,\text{nm}$ when water supersaturation decreases to $ss_{\text{w}} \approx 0.1\%$. This is the minimum level of $ss_{\text{w}}$ required to activate INP for immersion freezing.

For immersion freezing of dust particles, the parameterization of Ullrich et al. (2017) (U17-imm) (Table 1; Eq. 1) is based on heterogeneous ice nucleation experiments at the cloud chamber AIDA (Aerosol Interaction and Dynamics in the Atmosphere)

of the Karlsruhe Institute of Technology. The desert dust ground samples used in this study originated from different desert dust locations around the world (Saharan, Takla Makan, Canary Island, Israel). The parameterization quantifies the desert dust ice nucleation efficiency as a function of ice-nucleation-active surface-site density $n_{\text{s}}(T)$ and dust dry surface area concentration $S_{\text{d,dry}}$. If the CCN activated fraction is less than 50%, Eq. (1) for U17-imm needs to be scaled to be representative for the CCN activated $S_{\text{dry}}$ (Ullrich et al., 2017). In this work, we apply the U17-imm parameterization taking into consideration the total

$S_{\text{dry}}$.

Additionally, the parameterization of DeMott et al. (2015) (D15) (Table 1; Eq. 2) addresses the immersion and condensation freezing activity of natural mineral dust particles based on laboratory studies using the continuous flow diffusion chamber (CFDC) of the Colorado State University's (CSU) and field data from atmospheric measurements in Saharan dust layers. D15 quantifies $n_{\text{INP}}$ as a function of temperature and the total number concentration of dust particles with dry radii larger

than 250 nm ($n_{250,\text{d,dry}}$). We note here that the ambient values of measured $n_{\text{INP}}(p,T)$ need to be transferred in standard (std) pressure and temperature conditions ($n_{250,\text{d,dry}}(p_0, T_0, T)$) before the use of (Eq. 2).

For the deposition nucleation of dust particles, the parameterizations of Steinke et al. (2015) and Ullrich et al. (2017) (S15 and U17-dep, respectively) quantify the ice nucleation efficiency as a function of $S_{\text{d,dry}}$ and $n_{\text{s}}(T, S_{\text{ice}})$ with $S_{\text{ice}}$ the ice sat-

uration ratio. Both were based on AIDA laboratory studies, but they used different dust samples. U17-dep (Table 1; Eq. 3) was based on ground desert dust samples from Sahara, Takla Makan, Canary Island and Israel while S15 (Table 1; Eq. 4) was based on dust samples from Arizona, which were treated (washed, milled, treated with acid) and are much more ice active than natural desert dusts particles on average. Although S15 parameterization was based on "treated" dust samples which usu-
ally show an enhanced freezing efficiency, it is used in the NMME-DREAM model (Non-hydrostatic Mesoscale Model on E grid, Janjic et al. (2001); Dust REgional Atmospheric Model, Nickovic et al. (2001); Pérez et al. (2006)) for INP concentration estimations (Nickovic et al. , 2016). For this reason, it is included in this work.

For the ice activation of soot particles, Ullrich et al. (2017) introduced two parameterizations, one for immersion freezing (Table 1; Eq. 5) and a second one for deposition nucleation (Table 1; Eq. 6). Both were based on experiments at the AIDA
chamber with soot samples generated from four different devices and quantify the soot ice nucleation efficiency as a function of $S_{dry}$ and $n_s(T)$ (for immersion) and $n_s(T, S_{ice})$ (for deposition).

Finally, the global type-independent $n_{INP}$ parameterization of DeMott et al. (2010) (Table 1; Eq. 7), was based on field data collected during nine field campaigns (in Colorado, eastern Canada, Amazonia, Alaska, and Pacific Basin) and analyzed with the CFDC instrument of the CSU. As the majority of the samples used for D10 were non-desert continental aerosols,
this INP parameterization has been considered to be suitable for addressing the immersion and condensation freezing activity of mixtures of anthropogenic haze, biomass burning smoke, biological particles, soil and road dust (Mamouri and Ansmann, 2016). From here on these mixtures are addressed as continental aerosols.

The $n_{250,dry}$ and $S_{dry}$ used in all the aforementioned parameterizations are calculated from the lidar extinction profiles as described in Section 3.2 and shown in Figures A1 and A2 in the Apendix.

Figure 1 provides an indication of the relative differences of the observed $n_{INP}$ in nature for immersion (right) and deposition (left) modes and in relation with the different aerosol compositions by showing a summary of the different $n_{INP}$ parameterizations. Specifically, the plot shows the fraction of the ice-activated particles ($f_i = n_{INP}/n_{50,dry}$) for desert dust (dark blue, orange, red, light blue), continental (green) and soot (black). The particle concentrations used here, are derived assuming an extinction coefficient of 50 $Mm^{-1}$ for each of the different aerosol types (dust, continental, soot). The shaded areas take into
account a range of the extinction coefficient from 10 $Mm^{-1}$ (lower limit) to 200 $Mm^{-1}$ (upper limit). The error bars mark the cumulative error in $f_i$ that results from the uncertainty in the lidar observations and their conversion to mass concentration as well as from the errors in the respective parameterizations. An overview of the typical values and the uncertainties used for the error estimation in this study is provided in Table 2. The deposition nucleation estimations in the left panel of Figure 1 are provided for $ss_i = 1.15$ (solid lines) and $ss_i = (1.05, 1.1, 1.2, 1.3, 1.4)$ (dashed lines) to give a perspective on the range of
possible values. Note here that although the immersion parameterizations were obtained using measurements at the temperature ranges of [-30, -14]°C (U17-imm, dust), [-35, -21]°C (D15, dust), [-34, -18]°C (U17-imm, soot) and [-35, -9]°C (D10, continental), they are extrapolated herein to extend over the immersion-freezing temperature range (dashed part of the lines in the immersion mode chart).

Figure 1 (left panel) shows that, for deposition mode, the dust ice-activated fractions from S15 are several orders of magnitude
nitude higher than those of U17-imm (e.g. 4 orders of magnitude at -40°C and $ss_i = 1.15$%). Furthermore, the deposition

ice-activation fraction of dust and soot (from U17-dep) differ significantly with soot being more active than dust for $T < -38°C$ (up to 2 orders of magnitude) and dust being more active than soot for $T > -38°C$ (up to 4 orders of magnitude).

Figure 1 (right panel) shows that, for immersion mode, the dust ice-activated fractions obtained from D15 are one order of magnitude lower than those calculated with U17-imm. Laboratory ice nucleation measurements and corresponding instrument inter-comparisons, have shown that at a single temperature between two and four orders of magnitude differences are observed as a result of the natural variability of the INP active fraction (DeMott et al., 2010, 2017) or the use of different INP counters (Burkert-Kohn et al., 2017). Hereon, we consider D15 and U17-imm as the lower and upper bounds of the immersed $n_{INP}$ estimations for dust INP populations. Figure 1 (immersion mode panel) illustrates the dust activation increase of up to six orders of magnitude within the mixed-phase temperature regime ($-15\,°C$ to $-35\,°C$). For a $5\,°C$ decrease, $n_{d,INP}$ increases by about one order of magnitude. Moreover, we see that at T < -18°C the immersion freezing desert dust ice activation (D15) is higher than the continental one (D10) while this changes at T > -18°C. On the contrary, soot (U17-imm) has always lower $f_i$ than dust (from either D15 or U17-imm). The ice-activated fractions of continental (D10) and soot (U17-imm) aerosols have a relative difference that is always less than 60% at T < -18°C. At higher temperatures they diverge with continental $f_i$ to exceed the soot one by one order of magnitude at T > -11°C.

Additionally, Figure 1 provides an indication of the error induced at the lidar estimated $n_{INP}$ due to errors in the selected values of T and $ss_i$. The right panel shows that, for immersion mode a 5°C error in the assumed T can introduce an error of 1 order of magnitude in the dust related $n_{INP}$ estimations (U17-imm and D15) and 1/2 order of magnitude in the non-dust related estimations of D10. The same error (1/2 order of magnitude) is induced in the U17-imm(soot) (for T < -18°C). For deposition mode, a 5°C error in the assumed T can introduce an error of 1/2 order of magnitude in the dust related $n_{INP}$ estimations (U17-dep(dust) and S15). For the U17-dep(soot) estimates, and at T > - 45°C, the error in the assumed T has a significant impact in the $n_{INP}$ product (e.g. 1 order of magnitude between T = -45 and -40°C). On the contrary, at T < -45°C, the error in the assumed T has less impact in the final $n_{INP}$ product (between 100% and 200% for 5°C T error).

Regarding the deposition nucleation, a large variability of the onset saturation ratio is observed in laboratory experiments of different studies, with $ss_i$ varying for example at -40°C between 1 and 1.5 (Hoose and Möhler, 2012). In Figure 1 (left panel), we see the effect of the $ss_i$ on the estimated $n_{INP}$. In S15, $n_{INP}$ increase by 1 order of magnitude for 0.1 increase in the $ss_i$. In U17-dep(dust), 3 orders of magnitude $n_{INP}$ range is observed at -30°C for $ss_i$ between 1.05 and 1.4. The range is wider at lower temperatures (4 orders at -50°C). In U17-dep(soot) 4 orders of magnitude $n_{INP}$ range is observed at T < -40°C for $ss_i$ between 1.05 and 1.3. This variability provides an indication of the error induced in the lidar estimated $n_{INP}$ product due to the error in the selected $ss_i$. In the $n_{INP}$ profiles presented in Figure 11, $ss_i = 1.15$ is assumed (bold line here).

## 3 Instruments and methodology

The "INUIT-BACCHUS-ACTRIS" campaign in April 2016 was organized within the framework of the projects Ice Nuclei Research Unit (INUIT; https://www.ice-nuclei.de/the-inuit-project/), Impact of Biogenic versus Anthropogenic emissions on Clouds and Climate: towards a Holistic UnderStanding (BACCHUS; http://www.bacchus-env.eu/) and Aerosols, Clouds, and

Trace gases Research InfraStructure (ACTRIS; https://www.actris.eu/) and focused on aerosols, clouds and ice nucleation within dust-laden air over the Eastern Mediterranean. Although dust was the main component observed, other aerosol types were present as well such as soot and continental aerosols.

The atmospheric measurements conducted during the campaign included remote-sensing with aerosol lidar and sun photometers as well as in-situ particle sampling with two UAVs. The UAV provided observations of the INP abundance in the lower troposphere and they were operated from the airfield of the Cyprus Institute at Orounda (35°05'42"N, 33°04'53"E, 327 m asl, about 21 km west of Nicosia) (Schrod et al., 2017). An Aerosol Robotic Network (AERONET, Holben et al. 1998) sun photometer was located at the Cyprus Atmospheric Observatory of Agia Marina Xyliatou (35°02'19"N, 33°03'28"E, 532 m asl, 7 km west of the UAV airfield). Continuous ground-based lidar observations were performed at Nicosia (35°08'26"N, 33°22'52"E, 181 m asl) with the EARLINET PollyXT multi-wavelength Raman lidar of the National Observatory of Athens (NOA). For the second half of the campaign the lidar observations were complemented at Nicosia by a sun/lunar-photometer which was used to check the homogeneity of the aerosol loading between the different sites of Nicosia and Agia Marina.

## 3.1 Lidar measurements

The EARLINET PollyXT-NOA lidar measurements at 532 nm are used in this study for the derivation of particle optical properties and mass concentration profiles. Quicklooks of all PollyXT measurements can be found on the web page of PollyNet (Raman and polarization lidar network, http://polly.tropos.de). PollyXT operates using a Nd:YAG laser that emits light at 355, 532, and 1064 nm. The receiver features 12 channels that enable measurements of elastically (three channels) and Raman scattered light (387 and 607 channels for aerosols, 407 for water vapor) as well the depolarization of the incoming light at 355 and 532 nm. It also performs near-range measurements of two elastic and two Raman channels. More details about the instrument and its measurements are provided in Engelmann et al. (2016) and Baars et al. (2016). In brief, the nightime backscatter (b) and extinction (a) coefficient profiles at 532 nm are derived using the Raman method proposed by Ansmann et al. (1992). The volume and particle depolarization ratio profiles are derived using the methodologies described in Freudenthaler et al. (2009) and Freudenthaler (2016). The daytime backscatter and extinction coefficient profiles are derived using the Klett-Fernald method (Klett, 1981; Fernald , 1984), assuming a constant value for the lidar ratio (LR). The daytime Klett profiles in Section 4.1 were derived using a lidar ratio of 50 sr at 15th of April and of 40 sr at 5, 9, 21 and 22 of April and a vertical smoothing length using a sliding average of 232.5 m. The integrated extinction coefficient profiles calculated with these LRs agree well with the collocated AERONET aerosol optical depth (AOD) observations. The LR values also are in agreement with the nighttime Raman measurements indicating mixtures of dust and anthropogenic/continental particles at heights between 1 and 3 km. The 2D backscatter coefficient curtain for Figure 4 is calculated with the methodology described by Baars et al. (2017).

In this work we also use space-borne observations from the Cloud-Aerosol Lidar with Orthogonal Polarisation (CALIOP) on board the Cloud-Aerosol Lidar and Infrared Pathfinder Satellite Observations (CALIPSO) satellite (Winker et al., 2009). During the campaign period CALIPSO passed over Nicosia at a distance of 5 km on 5 and 21 April 2016. Here, we use the CALIPSO L2 Version 4 (V4) aerosol profile products of 21st of April 2016 and consider only quality-assured retrievals (Marinou et al., 2017; Tackett et al., 2018).

## 3.2 INP retrieval from lidar measurements

We calculated the $n_{INP}$ profiles from the lidar measurements by first separating the lidar backscatter profile in its dust and non-dust components using the aerosol-type separation technique introduced by Shimizu et al. (2004) and Tesche et al. (2009). For this method we consider a dust particle linear depolarization ratio of $\delta_d = 0.31\pm0.04$ (Freudenthaler et al. , 2009; Ansmann et al.,

2011a) and a non-dust particle linear depolarization ratio of $\delta_{nd} = 0.05\pm0.03$ (Müller et al., 2007; Groß et al., 2013; Baars et al., 2016; Haarig et al., 2017). The observed particle linear depolarization ratio in between these marginal values is therefore attributed to a mixture of the two aerosol types. The dust extinction coefficient ($\alpha_d$) is calculated using the mean LR of $45\pm11$ sr for dust transported to Cyprus (Nisantzi et al., 2015). For the non-dust component, the extinction coefficient ($\alpha_c$) is calculated using a LR of $50\pm25$ sr which is representative for non-desert continental mixtures (Mamouri and Ansmann, 2014; Baars et al.,

2016; Kim et al., 2018). The profiles of $n_{250,d,dry}$, $S_{d,dry}$, $n_{250,c,dry}$ and $S_{c,dry}$ are calculated from the extinction coefficient profiles using the POLIPHON algorithm (POlarization-LIdar PHOtometer Networking) and AERONET-based parameterizations proposed by Mamouri and Ansmann (2015, 2016). Table 3 provides an overview of the corresponding formulas used for the calculations. Weinzierl et al. (2009) showed that for dust environments the AERONET-derived values of $S_{dry}$ are about 95% of the total particle surface area concentration (including particles with radius $< 50$ nm). This assumption has been validated

against airborne in-situ observations of the particle size distribution during the Saharan Mineral Dust Experiment (SAMUM; Ansmann et al. (2011b)) in Morocco. The correlation drops to ∼0.85$\pm$0.10 for urban environments based on ground-based in-situ measurements of particle size distributions at the urban site of Leipzig (Mamouri and Ansmann, 2016).

     The uncertainty in the products (considering the initial errors provided in Table 2) are as follows: The estimated $n_{250,d,dry}$ uncertainty is 30% in well-detected dessert dust layers ($\delta_d = 0.3$), 37% in less pronounce aerosol layers ($\delta_d = 0.2$) and exceeds

94% in aerosol layers with low dust contribution ($\delta_d < 0.1$). The uncertainty of the estimated $S_{d,dry}$ values is 38% in well-detected dessert dust layers, 44% in less pronounce aerosol layers and exceeds 97% in aerosol layers with low dust contribution. The overall uncertainties of the combined (dust & continental) $n_{250,dry}$ and $S_{dry}$ values are between 20 - 40% and 30 - 50% respectively. The steps of the procedure for obtaining the profile of $n_{250,dry}$ and $S_{c,dry}$, as described here, are illustrated in an example in Figure 2. In this example, we use the PollyXT measurements at Nicosia between 1 and 2 UTC on 21 April 2016.

In the final step, the $n_{INP}$ profiles are estimated using the ice nuclei parameterizations presented in Section 2 (Eq. (1)-(7)). For these calculations we are using collocated modeled profiles of the pressure, temperature and humidity fields. Specifically, for the PollyXT-based $n_{INP}$ calculations we use hourly outputs from the Weather Research and Forecasting atmospheric model (WRF; Skamarock et al. (2008)) which is operational at the National Observatory of Athens at a mesoscale resolution of 12 x 12 km and 31 vertical levels (Solomos et al., 2015, 2018). Initial and boundary conditions for the atmospheric fields and the

sea surface temperature are taken from the National Center for Environmental Prediction (NCEP) global reanalysis at $1°x1°$ resolution. For the CALIPSO-bases $n_{INP}$ calculations we use the track-collocated meteorological profiles from the MERRA-2 model (Modern-Era Retrospective analysis for Research and Applications, Version 2) which are included in the CALIPSO V4 product (Kar et al., 2018).

### 3.3 UAV in-situ measurements

Two fixed-wing UAVs, the "Cruiser" and the "Skywalker", performed aerosol measurements up to altitudes of 2.5 km agl (2.85 km asl). Both UAVs were used to collect INP samples onto silicon wafers using electrostatic precipitation. The Cruiser can carry a payload of up to 10 kg and it was equipped with the multi-INP sampler PEAC (programmable electrostatic aerosol collector) (Schrod et al., 2016). Skywalker X8 (a light UAV that can carry a payload of 2 kg) was equipped with a custom-built, lightweight version of a single-sampler PEAC (Schrod et al., 2017). In total, 42 UAV INP flights were performed to collect 52 samples during 19 measurement days: 7 Cruiser flights with a total of 17 samples during 6 days and 35 Skywalker flights with a total of 35 samples during 16 days.

The INP samples were subsequently analyzed with the FRIDGE INP counter (Schrod et al., 2016, 2017). FRIDGE is an isostatic diffusion chamber. The typical operation of FRIDGE allows for measurements at temperatures down to -30°C and relative humidity with respect to water ($RH_w$) up to water supersaturation. FRIDGE was originally designed to address the condensation and deposition freezing ice nucleation modes at water saturation and below. However, because condensation already begins at sub-saturation, its measurements at $RH_w$ between 95% and 100% encompass ice nucleation by deposition nucleation plus condensation/immersion freezing, which cannot be distinguished by this measurement technique. Recent measurements during a big-scale inter-comparison experiment with controlled laboratory settings showed, that the method compares well to other INP counters for various aerosol types (DeMott et al., 2018). However, sometimes FRIDGE measurements are on the lower end of observations when compared to instruments that encompass pure immersion freezing. The INP samples collected on 5, 15 and 21 April 2016 were used for comparison with the lidar-derived $n_{INP}$. The samples were analyzed at $-20\,°C$, $-25\,°C$ and $-30\,°C$ and at $RH_w$ of 95%, 97%, 99% and 101% with respect to water, or equivalently with respect to ice ($RH_{ice}$) 115% to 135% (Schrod et al., 2017). Hereon, the samples analyzed at $RH_w < 100\%$ are used as a reference for the deposition mode parameterizations and the samples analyzed at $RH_w$ of 101% are used as a reference for the immersion/condensation parameterizations. The errors of the INP measurements were estimated to be ∼20% considering the statistical reproducibility of an individual sample, for the samples analyzed for the experiment.

Cruiser was additionally equipped with an Optical Particle Counter (OPC, Met One Instruments, Model 212 Profiler) that measures the aerosol particle number concentration with 1 Hz resolution in eight channels ranging from 0.15 to 5 $\mu$m in radius (Mamali et al., 2018). The inlet of the OPC was preheated to keep the relative humidity below 50% to minimize the influence of water absorption. The Cruiser-OPC measurements on 5, 9, 15 and 22 April 2016 were used to calculate the $n_{250,dry}$ profiles discussed in Section 4.1.

The measurements from the OPC onboard the Cruiser UAV were validated at the ground, using a similar OPC and a Differential Mobility Analyzer (DMA). The first comparison showed underestimation for the bin with radius 1.5 $\mu$m to 2.5 $\mu$m and for the last bin with radius more than 5 $\mu$m. The second comparison showed that the OPC underestimates by less than 10% the number concentration of particles with radius between 0.15 $\mu$m and 0.5 $\mu$m (Burkart et al., 2010). Moreover, there are no data provided for particles with radius less than 0.15 $\mu$m. In order to correct for this under-sampling we fit a bimodal number size distribution on the in-situ data and derive a corrected $n_{250,dry}$ and $S_{dry}$. An example of this correction is shown in Figure 3 for

the number and surface size distributions measured at 1.2 km on 5 April 2016. For the cases discussed herein we found that the corrected $n_{250,\text{dry}}$ in-situ values were $\sim 20\%$ higher than the raw measurements.

## 3.4 Space-borne cloud observations

A-Train space-borne cloud observations are complimentary used to provide us the 3D distribution and characteristics of the clouds formed in the presence of the calculated $n_{\text{INP}}$. For the spatial distribution of the clouds formed during 21 April 2016, the true color observations from the MODIS instrument (Moderate Resolution Imaging Spectroradiometer) on board Aqua satellite are used. To get a better insight into the vertical cloud structure, we use outputs from the synergistic radar-lidar retrieval DARDAR (raDAR/liDAR; Delanoë and Hogan (2008)). The DARDAR retrieval (initiated by LATMOS and the University of Reading) uses collocated CloudSat, CALIPSO, and MODIS measurements and provides a cloud classification product (DARDAR-MASK; Ceccaldi et al. (2013)) and ice cloud retrieval products (DARDAR-CLOUD; Delanoë et al. (2014)) on a 60 m vertical and 1.1 km horizontal resolution (available at http://www.icare.univ-lille1.fr/projects/dardar). In this work, we use the DADAR-MASK product for cloud classification, and we utilize the DARDAR-CLOUD product to derive an estimation of the ice crystal number concentration ($n_{\text{ice}}$) of the scene. With increasing maximum diameter ($D_{\text{max}}$), the ice crystals become more complex and their effective density decreases (Heymsfield et al. , 2010). The DARDAR algorithm describe this relationship using a combination of in-situ measurements by Brown and Francis (1995) for low-density aggregates ($D_{\text{max}} > 300\,\mu m$) and by Mitchell (1996) for hexagonal columns ($D_{\text{max}} < 300\,\mu m$). We derive the $n_{\text{ice}}$ (DARDAR-Nice) following the approach presented by Sourdeval et al. (2018) on the DARDAR-Cloud parameters of the ice water content (IWC) and the normalization factor of the modified gamma size distribution ($N_0^\star$). The direct propagation of uncertainties for IWC and $N_0^\star$ provided by DARDAR-Cloud gives an estimate for the relative uncertainty in $n_{\text{ice}}$ from about $25\%$ in lidar-radar conditions to $50\%$ in lidar-only or radar-only conditions (Sourdeval et al. , 2018). This estimation accounts for instrumental errors and uncertainties associated with aprioris used in DARDAR-Cloud. In cases with high homogeneous nucleation rates or dominant aggregation processes, $N_i$ can be underestimated (respective overestimated) by additional $50\%$ due to deviations from the assumed particle size distribution. Due to further assumptions within DARDAR-Cloud (e.g. a fixed mass-dimensional relationship), additional uncertainties can increase the error of the retrieved $n_{\text{ice}}$. In Section 4.3, the retrieved $n_{\text{ice}}$ is only used as a hint to estimate the order of magnitude of the true $n_{\text{ice}}$.

## 4 Results and discussion

We present here the comparison between the UAV-OPC observations and the lidar-derived $n_{250}$ profiles (Section 4.1). The measurements used for this comparison corresponds to one intense dust event, where the UAV measurements were conducted under cloudy conditions (9 April) and three moderate dust/continental events, where the UAV flights were conducted under cloud-free conditions (5, 15 and 22 April). Subsequently, we present the comparison between the UAV-INP measurements and the lidar-derived $n_{\text{INP}}$ during three days with moderate dust load conditions (Section 4.2). From a total of six INP samples, one sample is collected during 21 of April in the presence of a pure dust event under cloudy conditions and the remaining five

samples are collected during 5 and 15 of April inside dust & continental aerosol layers under cloud-free conditions. A brief description of the aerosol conditions of the measurements used are provided herein.

On 5 April 2016, a homogeneous elevated dust layer was observed above the lidar station at 1.0-1.8 km from 0 to 8 UTC which was later on mixed into the developing planetary boundary layer (PBL). In the next hours (until 12 UTC), only moderate variability was observed above the station (in the lidar backscatter coefficient and $\delta_p$ curtains - not shown). The UAV samples were collected between 11:37 and 11:57 UTC at 30 km west of the lidar site with westerly winds prevailing. Constant $\delta_p$ of around 0.15 between 0.5 and 2.5 km supports the qualitative homogeneity between the two observation sites during this time period.

On 9 April 2016, a thick pure dust layer (with $\delta_p \approx 0.3$) was observed above the lidar station, as part of a major dust event above Cyprus between 8 to 11 April 2016. The mean AOD at Nicosia was 0.83 (at 500-nm) with a corresponding mean Ångström exponent of 0.17 (at 440-870 nm). During the event, ice and water clouds were frequently formed at the top of the dust layer (mainly between 3 and 6 km). DREAM model and backward trajectory analysis revealed that this event originated from the central Sahara, with the dust particles being advected by a southwesterly flow directly towards Cyprus, reaching the island after one day (Schrod et al., 2017). The UAV samples were collected between 8:12 and 8:23 UTC inside the dust layer and these observations were compared with the lidar-derived profiles at 6:50-6:59 UTC (a closer-in-time lidar\UAV collocation is not possible due to clouds with a cloud base at 4 km later on). The OPC concentrations collected that day were the highest observed during the period of the INUIT-BACCHUS-ACTRIS experiment.

On 15 April 2016 a persistent elevated dust layer was observed above Nicosia. Backward trajectory analysis (not shown) revealed that this dust event originated from Algeria and that the dust plume was transported over Greece and Turkey before reaching Cyprus. Cruiser UAVs collected samples between 6:54 and 8:45 UTC (during the boundary layer development). At that time, a pure dust layer ($\delta_p \approx 0.3$) was present between 2.5 and 3.8 km height. Below 2.0 km the dust was mixed with spherical/continental particles from the residual layer with $\delta_p$ decreasing with height (reaching $\sim 0.1$ at 0.6 km). During the 2-hour flight, the scene above the station changed considerably, with 31% increase in the aerosol optical thickness (from 0.33 to 0.48) and 16% decrease in the Ångström exponent (from 0.31 to 0.26). The UAV measurements that day reached heights of up to 2.2 km, thus capturing only the mixed bottom layer and the lower part of the elevated dust layer. For the comparison with the lidar-derived concentrations, only the UAV measurements inside the lower part of the elevated dust layer (1.7 - 2.2 km) are used.

The pure dust event on 20 to 21 April 2016 is considered the golden case of our dataset, as it has been observed simultaneously with the PollyXT lidar, the UAVs and the A-Train satellites. Additionally, it is the only pure-dust event of our dataset where we have simultaneously good lidar observations and in-situ INP measurements. Figure 4 provides an overview of the times and heights of the PollyXT measurements, along with the CALIPSO overpass and UAV measurement times, between 20 and 22 April 2016. During that period atmospheric conditions supported the transport of dust from the Saharan desert and the Arabian Peninsula to the Eastern Mediterranean ($\delta_p = 0.28 \pm 0.03$) (Floutsi , 2018). The elevated dust plume arrived over the lidar site at 4-5 km height ($\sim 15$ UTC on 20 April 2016), quickly widened to stretch from 2 to 8 km height with the top of the main plume at 5 km height, and disappeared at 18 UTC on 21st of April. On that day, ice clouds were formed within

the dust plume and were present between 02:00 and 10:45 UTC above Nicosia. As shown in the figure, UAV flights were performed inside the dust layer on 21 April 2016 (OPC measurements and INP sampling). The event was captured from the A-Train satellites at 11:01 UTC (CALIPSO over-pass time). Figure 5 provides an overview of the aerosol and clouds above the area, with the MODIS true color image (upper panel) and the combined DARDAR and CALIPSO L2 feature mask (lower panel). Dust is observed above the broader region in altitudes up to 6 km and ice clouds are formed inside the dust layer South of Cyprus in altitudes greater than 4 km (T < 0°C). The ice clouds are detected/characterized at 1 km horizontal resolution (DARDAR-MASK product) while the dust plume is detected at 20 and 80 km horizontal resolution (CALIPSO L2 product).

On 22 April 2016 a transported plume was detected between 03:00 and 10:00 UTC, in altitudes of 1 to 2 km above Cyprus. The layer consisted of a mixture of dust with pollution aerosol and is characterized by a homogeneous particle linear depolarisation ratio of $\delta_\mathrm{p} = 0.17 \pm 0.03$. UAV flights (OPC and INP sampling) were performed in the mixed layer during that day between 04:32 and 05:13 UTC (Figure 4).

All in-situ samples were collected at a location about 28 km to the west of the lidar site, thus the atmospheric homogeneity of the two areas had to be considered to select suitable measurement times for the comparisons. For this analysis we used the sun-photometer measurements at Agia Marina and Nicosia, backward trajectories, model fields and MODIS measurements. This was especially necessary for the case on 21 of April when clouds were formed at the top of the dust layer. During that day, the CALIPSO-derived $n_\mathrm{INP}$ at 11:01 UTC were compared to UAV-measured $n_\mathrm{INP}$ acquired approximately one and a half hours earlier (between 8:30 and 9:40 UTC). The space/time homogeneity of the CALIPSO-derived $s_\mathrm{dry}$ and $n_\mathrm{250,dry}$ profiles (acquired shortly after the end of the cloudy period) is confirmed by the respective estimates from the PollyXT measurements during 1 to 2 UTC (before the beginning of the cloud formation) as shown in Figure 6. The different measurement times of the ground-based and spaceborne lidars are marked in Figure 4. For the CALIPSO profiles, along-track observations ±80 km away from the lidar station are used. During that time, the dust plume declined by approximately 300 m. Nevertheless, CALIPSO and PollyXT retrieved profiles are in agreement within their error bars within the dense dust plume. Aerosol conditions were less homogeneous above and below this layer (see Figure 4) causing stronger differences between the profiles of the four parameters from the two instruments. The comparison between the CALIPSO-derived $n_\mathrm{INP}$ and the UAV measurements from this case are discussed in Section 4.2 (see Figure 9).

## 4.1 Evaluation of the $n_\mathrm{250,dry}$ retrieval

For the assessment of the lidar-based $n_\mathrm{250}$-retrieval we used the OPC measurements on 5, 9, 15 and 22 April. The profiles of $n_\mathrm{250,dry}$ retrieved from PollyXT observations and in-situ measurements are shown in Figure 7 (upper panel). The lidar dust-only profiles (orange lines) are calculated from the dust extinction profiles and Eq. 8 (Table 3). The remaining non-dust component is considered continental with $n_\mathrm{250,c,dry}$ provided by Eq. 10 (Table 3). The total $n_\mathrm{250,dry}$ profiles (Figure 7, upper panel, black lines) are the summation of $n_\mathrm{250,d,dry}$ and $n_\mathrm{250,c,dry}$. The red dots correspond to the uncorrected UAV $n_\mathrm{250,dry}$ measurements. The blue dots correspond to the corrected UAV $n_\mathrm{250,dry}$ measurements (as described in Section 3.3). We use only the respective height ranges at which homogeneous aerosol conditions allow for a comparison of the UAV- and lidar-derived estimates. These measurements correspond to heights above 0.5 km on 5th of April, above the PBL on 9 and 15 April (> 1 km and > 2 km

respectively) and above the nocturnal boundary layer on 22 April ($> 0.7$km). It seems that the distance has little impact on the lidar-derived and the in-situ measured $n_{250,\mathrm{dry}}$ presented in Figure 7, with most of the in-situ-derived $n_{250,\mathrm{dry}}$ being well within the error bars of the lidar retrieval when considering the contributions of both mineral dust and continental pollution. On 9 April we observed the highest differences between the lidar-derived and in-situ-measured $n_{250,\mathrm{dry}}$, which may be attributed to the $\sim$1 hr time difference between the in-situ sampling and the lidar retrieval (limitation due to mid-level clouds as discusses already). Nevertheless, the case is included here, as it represent the strongest dust event observed during the campaign. Overall, the values of $n_{250,\mathrm{dry}}$ varied between 1 and $50\,\mathrm{cm}^{-1}$.

Figure 8 provides a quantitative comparison of the observations presented in Figure 7 for lidar retrievals of $n_{250,\mathrm{dry}}$ considering both mineral dust and continental pollution and the corresponding in-situ measurements at the same height levels. Again, we see that the results agree well within the error bars of the lidar retrieval with $R^2 = 0.98$. The uncertainties of the UAV-derived $n_{250,\mathrm{dry}}$ values presented in Figure 7 and Figure 8 correspond to the standard deviation of the 30 seconds average (OPC initial resolution of 1 second). The error in the OPC data due to the assumption of the refractive index and the shape of the particles used for the derivation of the particle size distribution from the OPC measurements, were not taken into account in this study. Nevertheless, it is not expected to be high because the refractive index used is characteristic for dust particles (n=1.59). We have to keep in mind the effect of a possible inhomogeneity between the two stations. In view of all uncertainty sources, the lidar- and UAV- derived $n_{250,\mathrm{dry}}$ are in good agreement. In terms of absolute values, the lidar-derived $n_{250,\mathrm{dry}}$ are slightly lower than the UAV-derived ones. We conclude that lidar measurements are capable to provide reliable spatio-temporal distributions of $n_{250,\mathrm{dry}}$ in cases with dust and continental aerosol presence with an uncertainty of 20 to 40%.

The profiles of $S_{\mathrm{dry}}$ retrieved from PollyXT observations and in-situ measurements are shown in Figure 7 (lower panel). The dust-only profiles (orange lines) are calculated from the dust extinction profiles and Eq. 9 (Table 3). The remaining non-dust component is considered continental with $n_{250,\mathrm{c,dry}}$ provided by Eq. 11 (Table 3). The total $S_{\mathrm{dry}}$ profiles (Figure 7, lower panel, black lines) are the summation of $S_{\mathrm{d,dry}}$ and $S_{\mathrm{c,dry}}$. These profiles are compared to the total $S_{\mathrm{dry}}$ derived from the corrected in-situ number size distribution (e.g. Figure 3b). We see that the latter agree well within the uncertainty of the lidar-derived $S_{\mathrm{d,dry}}$ (orange line), but do not agree well when both mineral dust and continental pollution are considered (black line). This is mainly due to the sampling cut-off of the OPC instrument for particles with radius smaller than $150\,\mathrm{nm}$ which are mainly composed by the polluted continental particles. The effect is not seen in the corrected $n_{250}$, since the size ranges considered there are larger than $250\,\mathrm{nm}$.

## 4.2  Evaluation of the $n_{\mathrm{INP}}$ retrieval

For the assessment of the lidar-based $n_{\mathrm{INP}}$-retrieval, the UAV measurements on 5, 15 and 21 April 2016 are used. The samples of 5 and 15 of April were collected under the moderately mixed dust/continental conditions shown in Figure 7. On 5 April, the sample was collected at an altitude of $1.823\,\mathrm{km}$ altitude ($\delta_\mathrm{p} = 0.14 \pm 0.02$). On 15 April two samples were collected from $0.998\,\mathrm{km}$ and $1.281\,\mathrm{km}$ altitude ($\delta_\mathrm{p} = 0.15 \pm 0.03$). On 21 April, the pure-dust sample was collected from $2.55\,\mathrm{km}$ altitude ($\delta_\mathrm{p} = 0.28 \pm 0.03$) (Figure 4). Analysis performed in FRIDGE chamber provided the INP concentrations for these cases. The in-situ samples were analyzed at $-20\,^\circ\mathrm{C}$, $-25\,^\circ\mathrm{C}$ and $-30\,^\circ\mathrm{C}$. For the deposition nucleation (Figure 9a) and (Figure 10a), the

samples were analyzed at $RH_\mathrm{w}$ of 95%, 97%, and 99%, leading to three values of $S_\mathrm{ice}$ for each temperature (1.16, 1.18 and 1.23 for $-20\,^\circ$C, 1.21, 1.24 and 1.26 for $-25\,^\circ$C and 1.27, 1.30 and 1.33 for $-30\,^\circ$C). For the immersion freezing (Figure 9 b), the samples were analyzed at $RH_\mathrm{w}$ of 101%, leading to $S_\mathrm{ice}$ of 1.23, 1.29 and 1.35 for the temperatures of $-20\,^\circ$C $-25\,^\circ$C and $-30\,^\circ$C, respectively. For $T = -20\,^\circ$C, $RH_\mathrm{w} = 101\%$ and $S_\mathrm{ice} = 1.23$, we refer to the freezing process as condensation

freezing.

The sample of 21 April was analyzed by single particle analysis using a scanning electron microscope, which showed that 99% of the particles were dust and 1% was Ca sulfates and carbonaceous particles (Schrod et al., 2017). This sample is used in order to evaluate the performance of the $n_\mathrm{INP}$ lidar estimates in a pure dust case, where (i) the errors originating from the first step of our methodology (separation in dust and non-dust aerosol components) are small ($\sim 30\%$) and (ii) the uncertainties

induced from the D10 and U17-(soot) parameterizations are minimum. Figure 9 shows the $n_\mathrm{INP}$ on 21 April as they were calculated from the CALIPSO lidar measurements (colored symbols) and measured from the UAV-FRIDGE samples (black triangles), (a) for deposition nucleation (as a function of saturation over ice) and (b) for condensation and immersion freezing (as a function of temperature).

Likewise, we are using all the aforementioned cases, in order to evaluate the performance of the $n_\mathrm{INP}$ lidar estimates in cases

with dust and continental aerosols. Figure 10 shows scatter plots of all the lidar-estimated $n_\mathrm{INP}$ (from PollyXT and CALIPSO) against the in-situ measurements for (a) deposition nucleation and (b) condensation and immersion freezing. In Figure 10 (b) the ratio between the lidar-derived and the in-situ $n_\mathrm{INP}$ is provided as a function of temperature. Similar results are observed for both the pure dust (Figure 9) and the dust and continental cases (Figure 10), with the lidar estimated $n_\mathrm{INP}$ during the pure dust event to show the best agreement with the in-situ.

For the $n_\mathrm{INP}$ retrievals in the deposition mode we see that, using the U17-dep in a dust case the lidar-derived concentrations are in excellent agreement with the in-situ observations (well within their uncertainties), with $n_\mathrm{INP}$ values to span over 2.5 orders of magnitude (for different ice supersaturation conditions) and the retrievals to capture the whole extend of this range (Figure 9a). The lidar-retrieved U17-dep values in this case are dominated from the dust related $n_\mathrm{INP}$ (estimated from Eq. 3; Table 1), with the non-dust related $n_\mathrm{INP}$ (estimated from Eq. 6; Table 1) being five orders of magnitude lower. In dust and

continental cases (Figure 10a), the 97% of all the U17-dep lidar-derived $n_\mathrm{INP}$ are within the error bars of the in-situ and within a factor of 10 around the 1:1 line (r=0.75). The $n_\mathrm{INP}$ sampled with the UAVs ranged between 0.02 and $20\,\mathrm{L}^{-1}$. Using S15 parameterization, the predicted $n_\mathrm{INP}$ values are 3 to 5 orders of magnitude larger than the in-situ measurements in both dust and dust-continental cases (r=0.42). An overestimation was already expected as discussed in Section 2 and Steinke et al. (2015) but for completeness we include these results.

Figure 9 (b) and Figure 10 (b) shows the lidar derived immersion/condensation INPs. U17-imm dust-related $n_\mathrm{INP}$ are calculated using the INP parameterization of Eq. 1 (Table 1) with the $S_\mathrm{d,dry}$ from Eq. 9 (Table 3). The D15 dust-related $n_\mathrm{INP}$ are calculated using the Eq. 2 (Table 1) with the $n_{250,\mathrm{d,dry}}$ from Eq. 8 (Table 3). The D10 continental-related $n_\mathrm{INP}$ are calculated using the Eq. 7 (Table 1) with the $n_{250,\mathrm{c,dry}}$ from Eq. 10 (Table 3). The D15+D10 values for the total (dust + continental) aerosol in the scene, are the summation of the aforementioned D15 (dust-related) and D10 (continental-related) $n_\mathrm{INP}$ calculations (See

Figure A1 and A2 in Appendix). We did not include the U17-imm soot estimates in the plot since these are quite similar to the

estimated values from D10 at temperatures < -18°C (Section 2; Figure 1). Consequently, for the total INP load in the scene, the estimations provided from the D15+D10 are similar to the ones provided from D15+U17-imm(soot). In the rest of this manuscript, we will discuss only the joint D15+D10 estimates, keeping in mind that the same conclusions apply for the joint D15+U17-imm(soot) estimates.

5    In Figure 9 (b) and Figure 10 (b) we see that the lidar-derived $n_{INP}$ using D15 for dust and D10 for continental particles are in good agreement with the in-situ observations, within the respective uncertainties for the samples analyzed at −20°C and −25°C. The best $n_{INP}$ agreement is observed for the pure-dust sample analyzed under condensation freezing conditions (at −20°C): with in-situ measurements of $3.6 \pm 0.1\,\mathrm{L}^{-1}$ and lidar-derived D15+D10 estimates of $3.8\,\mathrm{L}^{-1}$. From them, $2.4\,\mathrm{L}^{-1}$ originated from the D15 dust contribution and the $1.4\,\mathrm{L}^{-1}$ from the D10 non-dust contribution (although the contribution from

10    the non-dust INP at lower temperatures was insignificant with non-dust concentrations of one order of magnitude lower than the dust ones). Using all the dust and continental cases we see that, for the samples analyzed under condensation freezing conditions, the D15+D10 estimated $n_{INP}$ are no more than 2.5 times higher than the in-situ measurements (Figure 10b). Larger differences are observed at the temperatures where immersion freezing dominates over condensation as the main INP pathway, with 1.5 - 7 times larger values at −25 °C and 4 - 13 times larger values at −30 °C. Indicatively, for the pure dust case, at T =

15    -25°C the in-situ $n_{INP}$ were $12\pm3\,\mathrm{L}^{-1}$ and the D15+D10 lidar-derived $n_{INP}$ were $26\,\mathrm{L}^{-1}$ (with a negative error of $14\,\mathrm{L}^{-1}$). At T = -30°C, the in-situ $n_{INP}$ were $62 \pm 14\,\mathrm{L}^{-1}$ while D15+D10 $n_{INP}$ estimates were one order of magnitude higher ($242\,\mathrm{L}^{-1}$). Overall, in 85% of the analyzed cases, the D15+D10 lidar retrievals are less than an order of magnitude higher than the UAV measurements. Regarding the U17-imm lidar-derived $n_{INP}$ values, they are overall 1 to 3 orders of magnitude higher than the in-situ ones. In particular they are 3-11, 2-80 and 2-1000 times larger than the samples analyzed at FRIDGE chamber at −20 °C,

20    −25 °C and −30 °C, respectively. Nevertheless, the in-situ observations are withing the uncertainty of the parameterization for all the cases. Indicatively, for the pure dust case, the U17-imm lidar-derived $n_{INP}$ values are $50\,\mathrm{L}^{-1}$ at T= -20°C. Recent comparisons of $n_{INP}$ derived from samples analyzed in FRIDGE chamber usually present good linear correlations but somewhat lower values with observations derived from pure immersion paths (e.g. D15) (DeMott et al., 2018). Possible reasons for these discrepancies may be (a) deficits and inadequacies in instrumentation and measurement techniques, (b) the lacking overlap of

25    the freezing modes, (c) inconsistencies between the inlet systems of the parameterization measurement (using cutoffs) and the in-situ measurements (using no cutoff) and (d) a variation in $RH_w$ (D15: 105%; FRIDGE: 101%) (Schrod et al., 2017).

The error bars of the lidar-based $n_{INP}$ estimations in Figure 9 and Figure 10 are calculated using Gaussian error propagation together with the typical uncertainties provided in Table 2. In DeMott et al. (2015), a standard deviation of two orders of magnitude is reported as the uncertainty of the D15 parameterization. In the same plots, the uncertainty of the $n_{INP}$ from

30    in-situ data is very low. Under most experimental conditions, the repeatability of the ice nucleation in the FRIDGE chamber dominates other uncertainties. An uncertainty of 20% has been suggested as a useful guideline for the uncertainty of the intrinsic measurements, corresponding to the statistical reproducibility of an individual sample. However, it has also been reported that natural variability by far outweighs the intrinsic uncertainty (Schrod et al., 2016). We need to consider the full uncertainty including precision and accuracy. The DeMott et al. (2018) inter-comparison of INP methods saw that at all temperatures

and for various test aerosols the $n_{INP}$ uncertainty for immersion freezing is one order of magnitude, while for deposition condensation the uncertainty is expected to be even larger.

Our analysis suggests that the D15+D10 (and D15+U17-imm(soot)) immersion/condensation parameterization (applicable for the temperature range -35 °C to $-9$ °C) and the U17-dep parameterization (applicable for the temperature range $-50$ °C to $-33$ °C) agree well with in-situ observations of $n_{INP}$ and can provide good $n_{INP}$ estimates in pure-dust and dust-continental environments. The U17-imm pure immersion parameterization provides 1-2 orders of magnitude larger values, we therefore consider the $n_{INP}$ estimates according to D15+D10 as the lower boundary of possible values, with the actual values to be up to one order of magnitude larger in the temperature regime of immersion freezing.

### 4.3   $n_{INP}$ profiles from PollyXT and CALIOP during the evolution of mixed-phase clouds in a Saharan dust event

The case study of 21 April 2016 demonstrates the feasibility of the proposed methodology to provide profiles of cloud-relevant aerosol parameters up to the cloud levels, using (ground-base and space-borne) lidar measurements. In particular for this case, the temporarily averaged PollyXT lidar observations at 1-2 UTC and the spatially averaged CALIPSO observations at 11:01 UTC provide us the information of the $n_{250,dry}$, $S_{dry}$ and $n_{INP}$ right before and after the cloud event which was formed inside the dust layer that day between 02:00 and 10:45 UTC. The profiles of $n_{250,dry}$ and $S_{dry}$ before (PollyXT) and after (CALIPSO) the cloud event are the ones already presented in Figure 6. As discussed above, the dust plume declined by approximately 300 m during that period while its $n_{INP}$ stayed relatively constant inside its dense part. Above the main dust layer the aerosol conditions were variable, with multiple thin layers present up to 8 km altitude only before the appearance of the clouds. Specifically, a contribution of non-dust/continental particles is observed between 5.6 and 8 km agl ($n_{250,dry} = 0.4 \pm 0.2$ cm$^{-3}$; Figure 6 (d)) and three thin dust layers are visible at 6.4, 6.8 and 7.8 km with dust $n_{250,dry}$ of 2.9, 1.5 and 2.0 cm$^{-3}$, respectively, and a local minimum at 7.55 km (0.01 cm$^{-3}$) (Figure 6 (c)). Figure 11 shows the $n_{INP}$ concentrations derived from the different parameterizations at altitudes between 3 and 8 km agl. From the WRF and MERRA-2 assimilations we see that T < -35 °C in heights up to 7.8 km agl, which indicate that the immersion freezing mechanism is dominant in this case and that the deposition nucleation mechanism is not significant.

Figure 11 (a) shows that before the cloud formation the non-dust aerosols contribute to a gradual increase of $n_{INP}$ per height from $0.04$ L$^{-1}$ (4.5 km; -10 °C) up to $0.4$ L$^{-1}$ (5.8 km; -20 °C) and 4 L$^{-1}$ (7.8 km; -34 °C) (based on D10). Using U17-imm for soot we derived the $n_{INP}$ for the relevant non-dust particles of $10^{-4}$ L$^{-1}$ (-10 °C), 0.04 L$^{-1}$ (-20 °C) and 8 L$^{-1}$ (-34 °C). Figure 11 (a) shows here again the relatively good agreement between the lidar-derived non-dust $n_{INP}$ using D10 and U17-imm parameterizations at T< -20 °C and their significant discrepancies at lower temperatures. The dust aerosols in the scene contribute to a gradual increase of $n_{INP}$ inside the main dust layer from $0.05$ L$^{-1}$ (4.5 km; -10 °C) to $0.4$ L$^{-1}$ (5.3 km; -14 °C). Then a decrease of one order of magnitude is observed up to 6 km (0.06 L$^{-1}$; -20 °C) at the top end of the main dust layer. Above this altitude, a wavy $n_{INP}$ profile is observed with local maximal at 6.5, 7.0 and 7.9 km of 2 L$^{-1}$ (-22 °C), 4 L$^{-1}$ (-25 °C) and 200 L$^{-1}$ (-33 °C). The aforementioned values correspond to D15 estimates. The U17-imm dust estimates are 60 L$^{-1}$ (-22 °C), 200 L$^{-1}$ (-25 °C) and 1000 L$^{-1}$ (-33 °C). Overall, 91% of the total $n_{INP}$ is attributed to dust aerosols (D15) and 9% to non-dust/continental aerosols (D10) at altitudes between 6.3-8 km (Temperatures < -21 °C). These abundances are reversed

inside the main dust layer (altitudes between 4-5.5 km; Temperatures [-20,-6] °C) where 34% of the total $n_{\text{INP}}$ is attributed to dust aerosols ($0.06\,\text{L}^{-1}$) and 66% to non-dust/continental aerosols ($0.12\,\text{L}^{-1}$). Shortly after the period analyzed here, mixed phase clouds are observed above Nicosia at first at altitudes between 5-7 km and during the rest of the cloudy period mainly above 4 km (Figure 4).

Figure 11 (b) show the lidar-derived $n_{\text{INP}}$ above the station shortly after the end of the cloudy conditions. At that time, the main dust layer is observed at altitudes up to 5.5 km without additional layers above it. These observations are close to the local noon with the air temperature above the station being increased by 2.7 degrees, leading to temperatures of 0 °C at 3.6 km and -15 °C at 5.4 km agl. At these altitudes, a relatively constant contribution of non-dust/continental particles is present ($n_{250,\text{dry}} = 0.4 \pm 0.2\,\text{cm}^{-3}$; Figure 6 (d)) which leads to a gradually increase of the non-dust $n_{\text{INP}}$ per height from $2\text{x}10^{-4}\,\text{L}^{-1}$

(4 km; -2 °C) to $10^{-2}\,\text{L}^{-1}$ (4.4 km; -5 °C) to $0.2\,\text{L}^{-1}$ (5.3 km; -12 °C) (D10 estimates). Additionally, the dust concentration per altitude is constant inside the dust layer and is decreased gradually above 4.6 km ($n_{250,\text{dry}} = 16\,\text{cm}^{-3}$; 4 - 4.6 km); Figure 6 (c)). The dust-related $n_{\text{INP}}$ per height are $8\text{x}10^{-3}\,\text{L}^{-1}$ (4 km; -2 °C), $3\text{x}10^{-3}\,\text{L}^{-1}$ (4.4 km; -5 °C) and $0.1\,\text{L}^{-1}$ (5.3 km; -12 °C) (D10 estimates). Overall, 25% of the total $n_{\text{INP}}$ is attributed to dust aerosols (D15) and 75% to non-dust/continental aerosols (D10) at altitudes between 3.8-5.6 km.

Taking into consideration all the aerosols, the $n_{\text{INP}}$ before and after the cloud development is $0.6\,\text{L}^{-1}$ and $0.1\,\text{L}^{-1}$ respectively at 5.3 km altitude (D15+D10 in Figure 6). This difference is due to the increase of the air temperature during the day and the decrease of $n_{250,\text{dry}}$ and $S_{\text{dry}}$. Before the cloud formation, the $n_{\text{INP}}$ values at [6,7.5] km are one order of magnitude larger than at 5.3 km ( $3\,\text{L}^{-1}$) and at 7.8 km two orders of magnitude higher than at 6 km ($200\,\text{L}^{-1}$). These results indicate that the particles in the main dust layer and the thin layers above it acted as seeding INPs for the cloud that formed in that layer,

affecting also its characteristics. However, further measurements are necessary to reach a more concrete conclusion, as for example, measurements of the atmosphere dynamics (e.g. from a wind lidar) and observations of the cloud evolution (e.g. from a cloud radar as in the recent study of Ansmann et al. (2019)). Although these measurements are absent from our ground-based instrumentation, we utilize the DARDAR-Nice product (based on the CLOUDSAT/CALIPSO observations on 21 April 2016 - Figure 5) as a hint for the true $n_{\text{INP}}$ of the scene, and we compare them with the neighboring CALIPSO $n_{\text{INP}}$ estimates.

Figure 12 shows the DARDAR $n_{\text{ice}}$ estimations along the A-train track (presented in Figure 5) and Figure 13 shows the $n_{\text{INP}}$ calculations on the same curtain using the D15+D10 (upper panel) and U17-imm (lower panel) parametrizations. Clouds are formed on top of the dust layer at latitudes of 32, 32.8 and 34 °N. The clouds observed at 32 and 32.8 °N are coupled/collocated with an aerosol layer at their cloud top, at altitudes of 6.3 and 7.3 km and temperatures of -18 and -25 °C respectively. Figure 14 shows the $n_{\text{ice}}$ profiles derived in these two clouds, along with the $n_{\text{INP}}$ profiles estimates in their vicinity. Due to the strong INP

number increase with deceasing temperature, the highest $n_{\text{INP}}$ concentrations are observed at the top of the upper aerosol-cloud layers. We assume that the ice crystals in these two clouds nucleate close to the cloud top (where the coldest temperatures are observed) and that afterwards the crystals grow and fall through the lower heights of the clouds formed. Moreover, we consider that no secondary ice production (SIP) processes are present in these clouds, or at least their contribution to the $n_{\text{ice}}$ is insignificant, as the cloud top temperatures are much lower that the temperatures where SIP have been observed (between

35  -3 and -8 °C) (Hallett and Mossop , 1974; Field et al., 2017; Sullivan et al., 2017, 2018). We compare the $n_{\text{INP}}$ at cloud top

height with the $n_{ice}$ inside the cloud, having in mind that, with our hypotheses, the $n_{ice}$ values can be up to the $n_{INP}$ values if all the INPs are activated to ice crystals. For the smaller cloud, at ∼32 °N, $n_{ice}$ between 0.8 and 8 L$^{-1}$ are retrieved and $n_{INP}$ between 0.3 to 2 L$^{-1}$ and 4 to 20 L$^{-1}$ are estimated with the D15+D10 and the U17-imm respectively. For the cloud at ∼32.8 °N, $n_{ice}$ between 0.4 and 60 L$^{-1}$ are retrieved and $n_{INP}$ between 3 to 20 L$^{-1}$ and 100 to 400 L$^{-1}$ are estimated with the D15+D10 and the U17-imm respectively. Overall, in these two clouds the $n_{INP}$ estimates in the top of the clouds have 1-2 order of magnitude uncertainty in their estsimates and one order of magnitude differences in the retrievals between each other. Additionally the retrieved DARDAR profiles provide us only with a hint of the order of magnitude of the true $n_{ice}$. Nevertheless the $n_{ice}$ estimates are between the estimated $n_{INP}$ values and within the errors of the two parameterizations. These results strengthen our conclusion that we can use the lidar-derived $n_{INP}$ from D15+D10 and U17-imm to estimate a minimum and maximum boundary of the $n_{ice}$ in a cloud formed in their presence, when immersion is the dominant mechanism.

## 5   Summary and conclusions

We present a methodology for deriving $n_{INP}$ profiles from lidar measurements and a comparison with in-situ UAV measurements of $n_{INP}$. More specifically, seven INP parameterizations are tested to obtain lidar (ground-based and space-borne) $n_{INP}$ estimates representative of mineral dust and continental/pollution/soot aerosol. We prove that a compilation of the parameterizations of DeMott et al. (2015) (D15) and DeMott et al. (2010) (D10), for dust and non dust particles respectively, is in good agreement with airborne in-situ measurements (Schrod et al., 2017) for addressing immersion/condensation freezing (at T>−35 °C). A similar conclusion is derived from the compilation of the parameterizations of DeMott et al. (2015) (D15) for dust and Ullrich et al. (2017) (U17) for soot. Specifically, lidar-derived $n_{INP}$ using D15+D10 (and D15+U17-imm(soot)) agree with the in-situ measurements within the reported uncertainty range of the D15 parameterization (i.e., two orders of magnitude; DeMott et al. (2015)). The best assessment for the deposition-related INPs is derived with the Ullrich et al. (2017) deposition nucleation parameterization for dust and soot (for T<−33 °C), with results agreeing with the UAV-FRIDGE measurements within one order of magnitude for different values of ice supersaturation.

The cloud-relevant aerosol parameters necessary for INP estimations ($n_{250,dry}$ and $S_{dry}$) are derived from lidar measurements as shown by Mamouri and Ansmann (2015, 2016). The comparison between the lidar-derived concentrations of dry particles with radii larger than 250 nm with coincident UAV-OPC in-situ measurements showed a good agreement with slightly lower values (32%) for the $n_{250,dry}$ derived by the lidar. This effect is less pronounced at low concentrations with squared correlation coefficient of 0.98. For the majority of the cases, we find that in-situ observations and remote-sensing estimates are in good agreement within their uncertainty ranges.

A further step for improving the lidar-derived INP retrievals and investigating the different parametrizations used is by conducting dedicated studies with collocated lidar measurements and additional temperature and humidity profiling in order to calculate the INP concentrations at real conditions, and the combination of the retrieved $n_{INP}$ with airborne in-situ ice concentration measurements.

Our methodology is validated for cases with dust presence. Additional measurements are required in order to define the optimum INP parameterizations for non-dust atmospheric conditions (e.g. continental, marine, smoke). Future experimental INP campaigns with airborne in-situ observations from aircrafts (including UAVs) collocated with lidar measurements at pure marine conditions and at mixed aerosol conditions could provide an ideal set-up for an in-depth investigation of the potential
of the lidar-based INP profiles in complex and non-dust atmospheric conditions.

The results presented in this study give us confidence to proceed to the next step which is to combine cloud-relevant lidar aerosol and wind parameters and cloud radar height-resolved observations to monitor the evolution of clouds embedded in aerosol layers. This will provide a unique opportunity to better understand aerosol-cloud-interactions in the field of heterogeneous ice formation.

Moreover, the study enhances the confidence for the production of global 3D products of $n_{250,\mathrm{dry}}$, $S_{\mathrm{dry}}$ and $n_{\mathrm{INP}}$ from the CALIPSO dataset. The application of our methodology to more than a decade-long CALIPSO measurements could provide valuable insight into global height-resolved distribution of $n_{250,\mathrm{dry}}$ and $n_{\mathrm{INP}}$ related to mineral dust, and possibly other aerosol types. This will enable global-wide studies of aerosol cloud interactions to combine the new product with satellite radar observations (CloudSat) and the upcoming EarthCARE (Earth Cloud Aerosol and Radiation Explorer) mission.

A challenge of a new global INP climatology will be the assessment of its underestimation at high altitudes where is known that CALIPSO observations can miss thin layers with small concentrations. A way to investigate the effect of the satellite-undetected layers in the $n_{250,\mathrm{dry}}$, $S_{\mathrm{dry}}$ and $n_{\mathrm{INP}}$ CALIPSO products is the utilization of ground-based lidar network observations as for example EARLINET and PollyNet.

*Data availability.* The CALIPSO and DARDAR data used in this study can be accessed through the ICARE Data and Services center:
http://www.icare.univ-lille1.fr/archive?dir=CALIOP/ and http://www.icare.univ-lille1.fr/archive?dir=MULTI_SENSOR/ respectively. The data is accessible after a free registration. The MODIS true color images can be accessed through the NASA Worldview center: https://worldview.earthdata. The in-situ INP data used in this study can be accessed through the BACCHUS database of INP observations. The database is accessible to members only but membership is free: http://www.bacchus-env.eu/in/. The in-situ INP dataset of this study can be reached through this link: http://www.bacchus-env.eu/in/info.php?id=72. The PollyXT lidar data, the in-situ OPC measurement and the WRF modeled profiles above
NICOSIA that are used in this study are available at: https://react-cloud.space.noa.gr/papers/. All data sets created during the calculation of the lidar-based number concentrations and the correction of the in-situ OPC number concentrations are provided upon request.

## Appendix A: Lidar retrievals of $n_{\mathrm{INP}}$

### A1   Methodological diagram for the analysis of the ground-based lidar measurements

The Figure A1 illustrates the general idea of the methodology followed for the INP estimations from the PollyXT measure-
ments. The equations for the conversions of the measured optical properties into the microphysical properties are provided in Table 3. The equations for the conversions of the microphysical properties to INPs are provided in Table 1.

## A2 Methodological diagram for the analysis of the space-borne lidar measurements

The Figure A2 illustrates the general idea of the methodology followed for the INP estimations from the CALIPSO measurements. The equations for the conversions of the measured optical properties into the microphysical properties are provided in Table 3. The equations for the conversions of the microphysical properties to INPs are provided in Table 1.

5  *Author contributions.*  V.A. and E.M. conceived the presented idea. E.M. performed the analysis, drafted the manuscript and designed the figures with support from M.T. and A.T.. A.N. guide and supervised E.M. on the ice nuclei mechanism and parameterizations and encouraged E.M. to investigate the errors of the lidar-derived INP concentrations. A.A. guided and supervised E.M. on the methodology and parameterizations for the lidar-derived INP concentrations and encouraged E.M. to investigate the errors of the in-situ OPC concentrations and the DARDAR $n_{ice}$ concentrations. J. Sch. performed the analysis of the INP samples and guided E.M. on the limitations of these measurements.

10  J.Sci. conceived and planned the experiment. M.P. was responsible for the UAV flights and the analysis of the OPC measurements. D.M. and A.T. performed the correction of the OPC measurements. R.E. supported essentially the PollyXT measurements during the campaign. H.B. supported the derivation of the PollyXT lidar backscatter coefficients. S.S. derived the WRF model profiles. F.E. derived the $n_{ice}$ from DARDAR. All authors provided critical feedback and helped shape the research, analysis and manuscript.

*Competing interests.*  The authors declare that they have no conflict of interest.

*Disclaimer.*  This article is part of the special issue "EARLINET aerosol profiling: contributions to atmospheric and climate research". It is not associated with a conference.

*Acknowledgements.*  We are grateful to Prof. Dr. Balis Dimitrios, Dr. Bingemer Heinz G. and Dr. Biskos George for their helpful contribution and advice on the interpretation of the results of this study. We thank Dr. Philippe Goloub for the provision of one of the cimel instruments operating during the campaign. We are grateful to the Cyprus Institute Unmanned System Research Laboratory (USRL) team

for their support in the operation of the UAV flights. We thank EARLINET (www.earlinet.org), ACTRIS (/www.actris.eu), AERONET (https://aeronet.gsfc.nasa.gov/) and AERONET-Europe for the data collection, calibration, processing and dissemination. We thank PollyNET group, and especially Dr. Dietrich Althausen, and Dr. Birgit Heese, for their support during the development and operation of the PollyXT lidar of NOA. We are grateful to the AERIS/ICARE Data and Services Center for generating and storing the DARDAR products and for providing access to the CALIPSO data used and their computational center (http://www.icare.univ-lille1.fr/). We thank the NASA

CloudSat Project and NASA/LaRC/ASDC for making available the CloudSat and CALIPSO products, respectively, which are used to build the synergetic DARDAR products. The provision of the HYSPLIT transport and dispersion model from the NOAA Air Resources Laboratory is also gratefully acknowledged. The research leading to these results has received funding from the European Research Council (ERC) project D-TECT (Does dust triboelectrification affect our climate?) under grant agreement no. 725698, European Union's Seventh

Framework programme (FP7/2007-2013) project BACCHUS (Impact of Biogenic versus Anthropogenic emissions on Clouds and Climate: towards a Holistic UnderStanding) under grant agreement no. 603445, the Deutsche Forschungsgemeinschaft (DFG) under the Research Unit FOR 1525 (INUIT) and the European Union's Horizon 2020 research and innovation program ACTRIS-2 (Aerosols, Clouds and Trace gases Research InfraStructure Network) under grant agreement no. 654109. Marinou E. acknowledges the support of the Deutscher Akademischer Austauschdienst (DAAD) through a post-doc scholarship (no. 57370121). The authors affiliated to National Observatory of Athens acknowledge support through the Stavros Niarchos Foundation. Voudouri K.A acknowledges the support of the General Secretariat for Research and Technology (GSRT) and the Hellenic Foundation for Research and Innovation (HFRI) (no. 294).

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

**Table 1.** Overview of INP parameterizations used in this study together with the freezing mode and the temperature range for which they have been developed. The parameterizations of D15 and U17-imm have been extrapolated to the temperature range from $-36\,°C$ to $-1\,°C$. In the equations, $n_{250,\mathrm{dry}}$ is in $\mathrm{cm}^{-3}$, $n_{\mathrm{INP}}$ in $\mathrm{L}^{-1}$, $T(z)$ in $K$ and $P$ in $hPa$. $p_0$ and $T_0$ hold for standard pressure and temperature.

| Parameterization name | Reference | Mode | T (°C) | Parameterization, $n_{\mathrm{INP}} =$ | Eq. |
|---|---|---|---|---|---|
| Dust: | | | | | |
| U17-imm | Ullrich et al. (2017) | immersion | -30 to -14 | $S_{\mathrm{d,dry}}n_{\mathrm{s}}(T)$ | (1) |
| | | | | with $n_{\mathrm{s}}(T) = \exp\left[150.577 - 0.517T\right]$ | |
| D15 | DeMott et al. (2015) | immersion condensation | -35 to -21 | $[n_{250,\mathrm{d,dry}}(p_0, T_0)^{[a_1(273.16-T)+b_1]}\exp\left[c_1(273.16-T)+d_1\right]](T_0 p)/(T p_0)$ | (2) |
| | | | | with $a_1 = 0.0$, $b_1 = 1.25$, $c_1 = 0.46$, $d_1 = -11.6$ | |
| U17-dep | Ullrich et al. (2017) | deposition | -67 to -33 | $S_{\mathrm{d,dry}}n_{\mathrm{s}}(T, S_{\mathrm{ice}})$ | (3) |
| | | | | with $n_{\mathrm{s}}(T, S_{\mathrm{ice}}) = \exp\left[a_2(S_{\mathrm{ice}}-1)^{\frac{1}{4}}\cos\left[b_2(T-\gamma_2)\right]^2 \mathrm{arccot}[\kappa_2(T-\lambda_2)]/\pi\right]$ | |
| | | | | and $a_2 = 285.692$, $b_2 = 0.017$, $\gamma_2 = 256.692$, $\kappa_2 = 0.080$, $\lambda_2 = 200.745$ | |
| S15 | Steinke et al. (2015) | deposition | -53 to -20 | $S_{\mathrm{d,dry}}n_{\mathrm{s}}(T)$ | (4) |
| | | | | with $n_{\mathrm{s}}(T) = 1.88 \times 10^5 \exp\left(0.2659\,\chi(T, S_{\mathrm{ice}})\right)$ | |
| | | | | and $\chi(T, S_{\mathrm{ice}}) = -(T - 273.2) + (S_{\mathrm{ice}} - 1) \times 100$ | |
| Soot: | | | | | |
| U17-imm | Ullrich et al. (2017) | immersion | -34 to -18 | $S_{\mathrm{c,dry}}n_{\mathrm{s}}(T)$ | (5) |
| | | | | with $n_{\mathrm{s}}(T) = 7.463\exp\left[-0.0101(T-273.15)^2 - 0.8525(T-273.15)+0.7667\right]$ | |
| U17-dep | Ullrich et al. (2017) | deposition | -78 to -38 | $S_{\mathrm{c,dry}}n_{\mathrm{s}}(T, S_{\mathrm{ice}})$ | (6) |
| | | | | with $n_{\mathrm{s}}(T, S_{\mathrm{ice}}) = \exp\left[a_3(S_{\mathrm{ice}}-1)^{\frac{1}{4}}\cos\left[b_3(T-\gamma_3)\right]^2 \mathrm{arccot}[\kappa_3(T-\lambda_3)]/\pi\right]$ | |
| | | | | and $a_3 = 46.021$, $b_3 = 0.011$, $\gamma_3 = 248.560$, $\kappa_3 = 0.148$, $\lambda_3 = 237.570$ | |
| Non-dust: | | | | | |
| D10 | DeMott et al. (2010) | immersion condensation | -35 to -9 | $[a_4(273.16-T)^{b_4}n_{250,\mathrm{c,dry}}(p_0, T_0)^{[c_4(273.16-T)+d_4]}](T_0 p)/(T p_0)$ | (7) |
| | | | | with $a_4 = 0.0000594$, $b_4 = 3.33$, $c_4 = 0.0265$, $d_4 = 0.0033$ | |

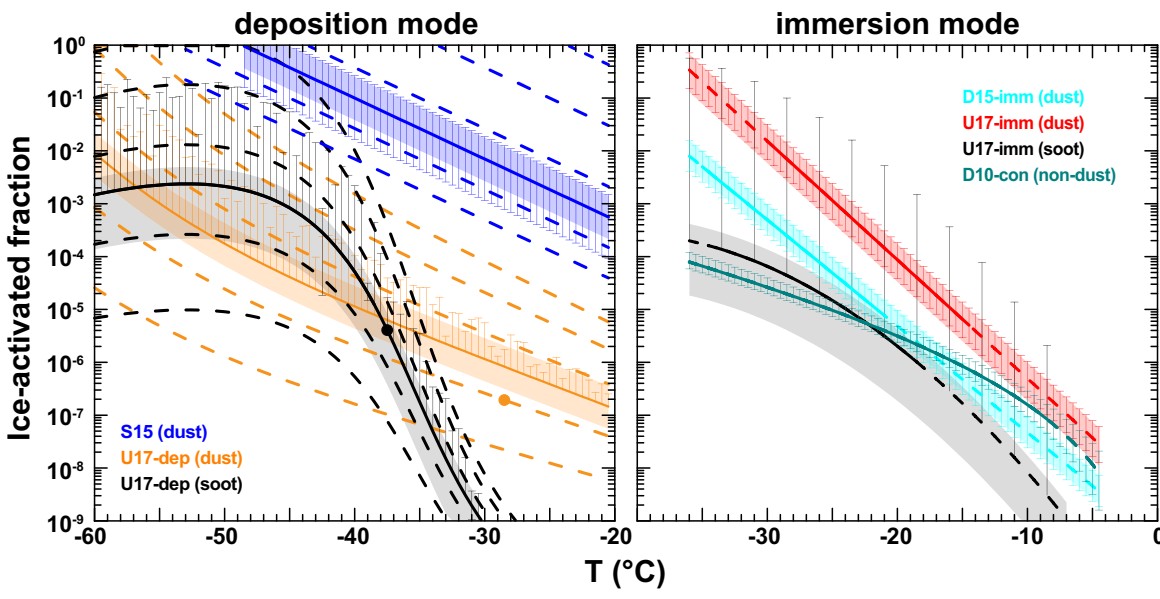

**Figure 1.** Fraction of ice activated particles for the deposition nucleation (left) and immersion freezing (right) parameterisations used in this study. The particle concentrations used are derived assuming an extinction coefficient of $50 \, \mathrm{Mm^{-1}}$ for each of the different aerosol types (dust, continental, soot). The shaded areas take into account a range of the extinction coefficient from $10 \, \mathrm{Mm^{-1}}$ (lower limit) to $200 \, \mathrm{Mm^{-1}}$ (upper limit). The error bars mark the error of the respective parameterisations from error propagation using the uncertainties provided in Table 2. Negative error bars that exceed the scale are not shown. In the deposition mode (left) panel, the bold lines correspond to ice supersaturation of 1.15 and the dashed lines to ice supersaturation of 1.05, 1.1, 1.2, 1.3 and 1.4. The black and orange dots indicate the maximum temperatures for which the parameterizations have been developed. In the immersion mode (right) panel, the parameterizations are extrapolated over the immersion-freezing temperature range (dashed lines).

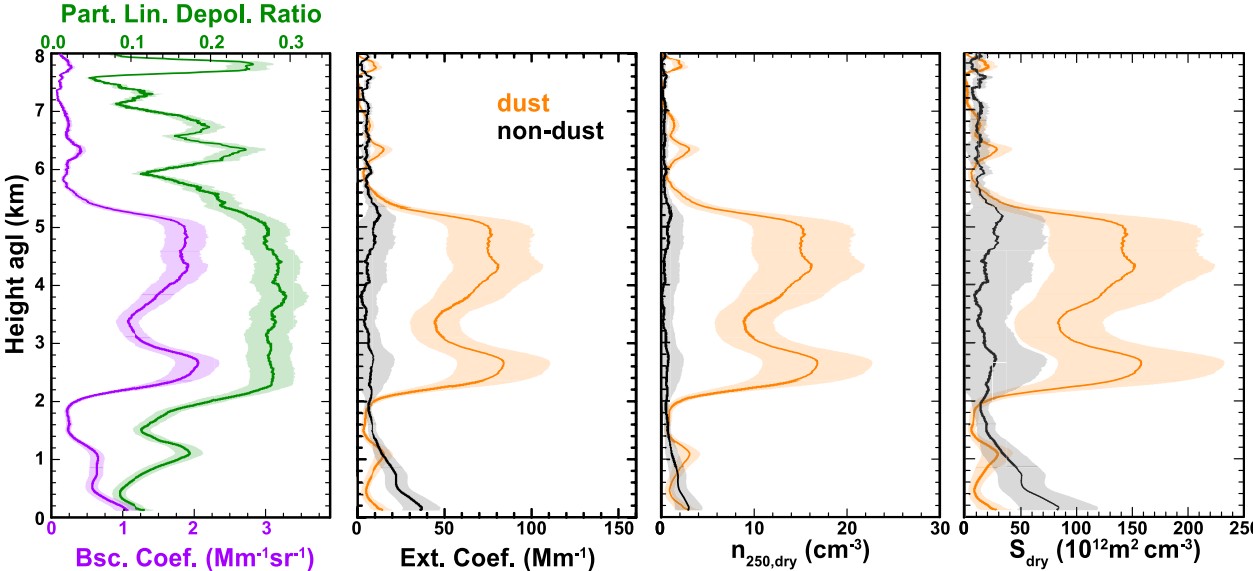

**Figure 2.** PollyXT profiles of the total particle backscatter coefficient (purple) and particle linear depolarisation ratio (green) measured between 1 and 2 UTC on 21 April 2016. The extinction coefficient as well as the number and surface concentration of particles with a dry radius larger than 250 nm related to mineral dust (orange) and non-dust aerosol (black) was obtained following the methodology described in Section 3.2.

**Table 2.** Values and typical uncertainties used for the estimation of $f_i$, $\alpha_d$, $\alpha_c$, $S_{d,dry}$, $S_{c,dry}$, $n_{250,d,dry}$, $n_{250,c,dry}$ and $n_{INP}$.

| Parameter | Value | Reference |
|---|---|---|
| $\beta_p$ | $0.15\,\beta_p$ | |
| $\alpha_p$ | $0.2\,\alpha_p$ | (only for $f_i$ estimations) |
| $\delta_p$ | $0.15\,\delta_p$ | |
| $\delta_d$ | $0.31 \pm 0.04$ | Freudenthaler et al. (2009); Ansmann et al. (2011a) |
| $\delta_{nd}$ | $0.05 \pm 0.03$ | Müller et al. (2007); Groß et al. (2013); Baars et al. (2016); Haarig et al. (2017) |
| $S_d$ | $45 \pm 11$ sr | Nisantzi et al. (2015) |
| $S_c$ | $50 \pm 25$ sr | Baars et al. (2016) |
| $c_{250,d}$ | $0.20 \pm 0.03\,\mathrm{Mm\,cm^{-3}}$ | Mamouri and Ansmann (2016) (Cape Verde, Barbados, Germany) |
| $c_{s,d}$ | $(1.94 \pm 0.68)\,10^{-12}\,\mathrm{Mm\,m^2\,cm^{-3}}$ | Mamouri and Ansmann (2016) (Cape Verde, Barbados) |
| $c_{290,c}$ | $0.10 \pm 0.04\,\mathrm{Mm\,cm^{-3}}$ | Mamouri and Ansmann (2016) (Germany) |
| $c_{s,c}$ | $(2.80 \pm 0.89)\,10^{-12}\,\mathrm{Mm\,m^2\,cm^{-3}}$ | Mamouri and Ansmann (2016) (Germany) |
| $\delta_T$ | 2 K | DeMott et al. (2017) |
| $S_{ice}$ | $1.15 \pm 0.05 S_{ice}$ | DeMott et al. (2017) |

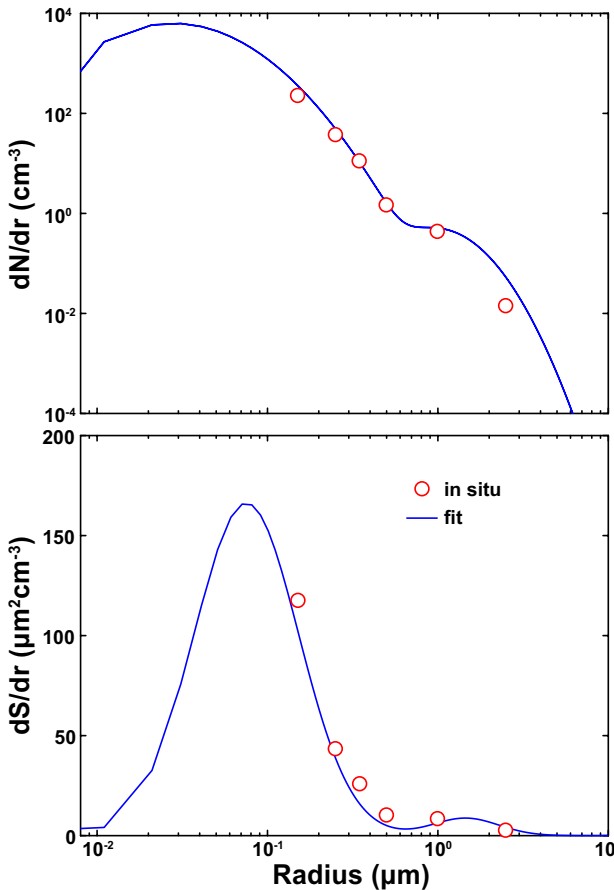

**Figure 3.** (a) The number size distribution used for the estimation of the corrected $n_{250,dry}$ (number concentration of particles with radius larger than 250 nm) and (b) the corresponding surface size distribution used for the estimation of the corrected $S_{dry}$ (surface concentration of all particles). In-situ measurements are denoted by red circles while the blue lines give the bimodal log-normal fit on the measurements. The example refers to the UAV-OPC data acquired at 1.2 km at 1045 UTC on 5 April 2016 (see Figure 7).

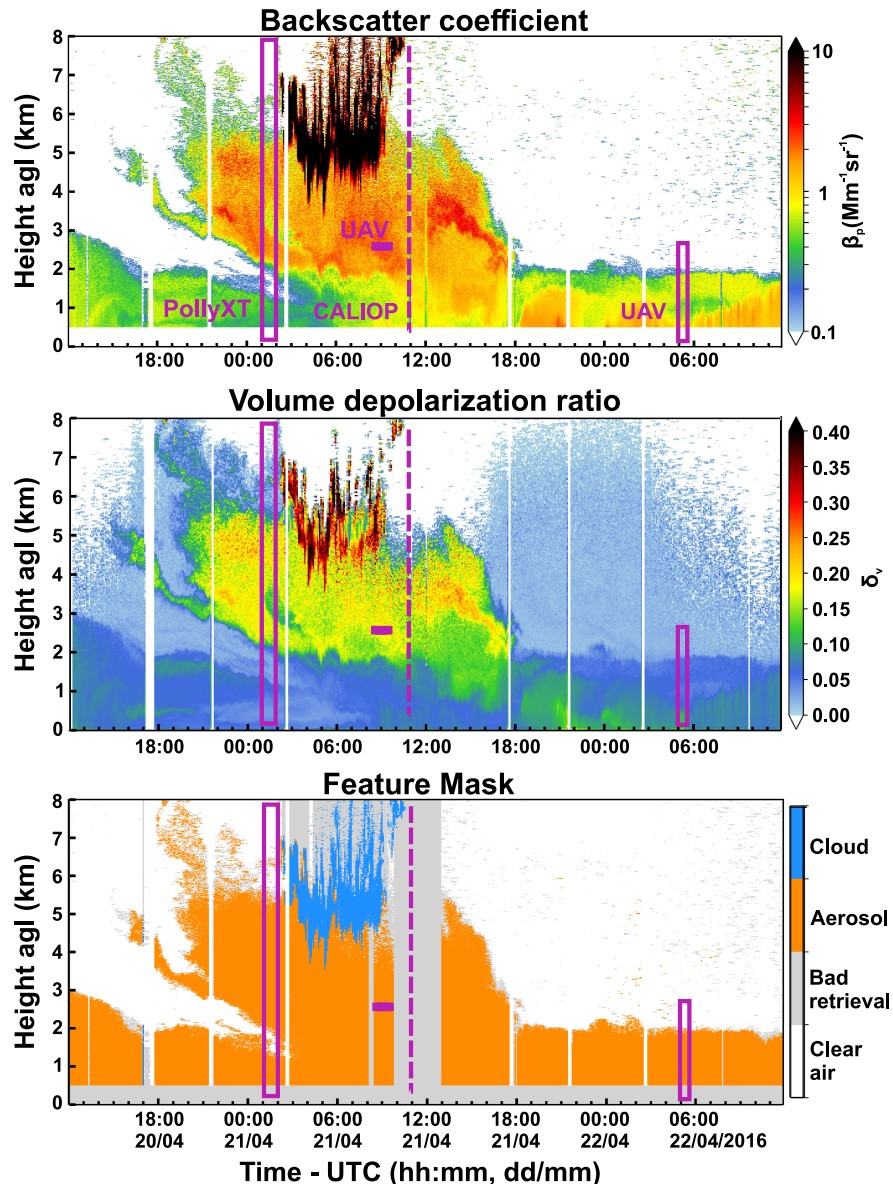

**Figure 4.** Time-height PollyXT observations between 12 UTC on 20 April 2016 and 12 UTC on 22 April 2016 of the backscatter coefficient at 1064-nm (up), the volume linear depolarization ratio at 532-nm (center) and the feature mask (bottom). The magenta markers refer to the analysed period of PollyXT (left box: 1 - 2 UTC on 21 April 2016), CALIOP (dashed line: 11:01 UTC on 21 April 2016) and UAV (horizontal bar: INP sampling between 8:30 and 9:40 UTC on 21 April 2016, right box: OPC measurements between 5:00 and 5:30 UTC on 22 April 2016) that are being referred to in this study. The bad retrievals in the feature mask refers to observations affected from (i) total attenuation due to clouds (ii) low signal-to-noise ratio and (iii) incomplete overlap.

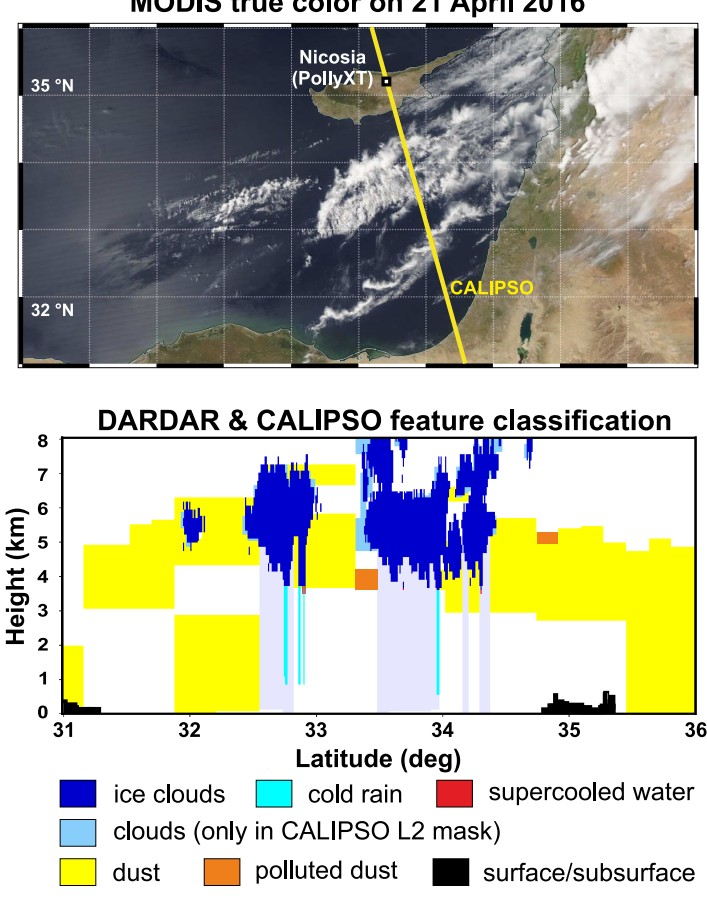

**Figure 5.** A-Train observations on 21 April 2016 at 11 UTC of MODIS-Aqua true color (up) and DARDAR & CALIPSO feature classification (bottom).

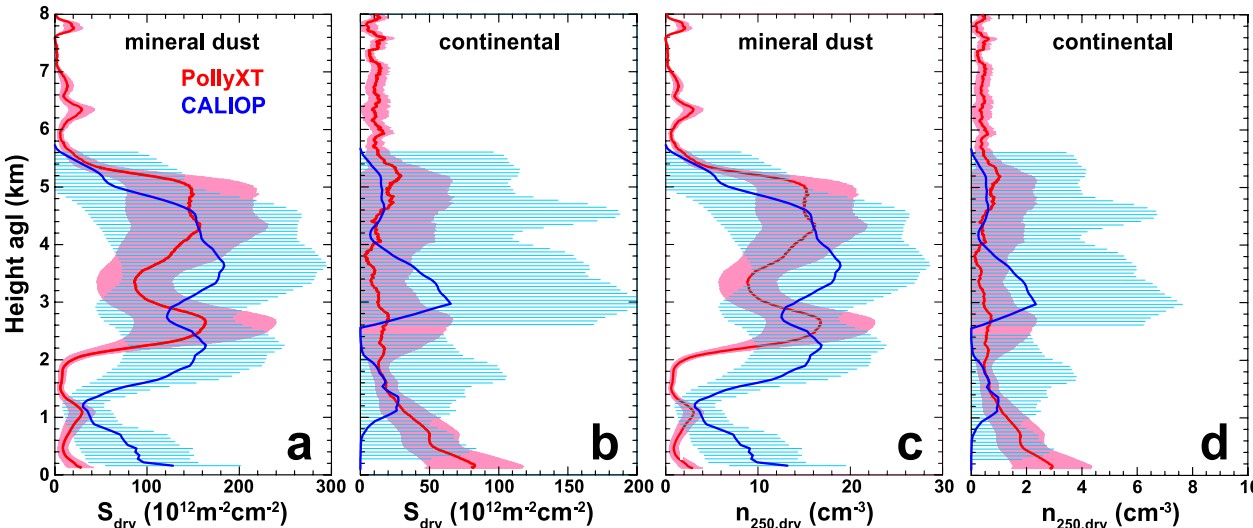

**Figure 6.** Profiles of the surface (a, b) and number concentrations (c, d) of mineral dust (a, c) and continental particles (b, d) with a dry radius larger than 250 nm derived from measurements with PollyXT between 1 and 2 UTC on 21 April 2016 (red) and retrieved from averaging 160 km of CALIOP measurements centred around an overpass at a distance of 5 km from Nicosia at 11:01 UTC on 21 April 2016 (blue).

**Table 3.** Overview of the AERONET-based parameterizations used in this study for the conversion of the measured optical aerosol properties ($\alpha_d$, $\alpha_c$) into the microphysical properties ($n_{250,d,dry}$, $S_{d,dry}$, $n_{250,c,dry}$ and $S_{c,dry}$). The parameterizations were introduced in Mamouri and Ansmann (2016). In the equations, $\alpha$ is in $\mathrm{Mm}^{-1}$, $c_{250}$ in $\mathrm{Mm\ cm}^{-3}$, $c_s$ in $\mathrm{Mm\ m}^2\ \mathrm{cm}^{-3}$, $n_{250,dry}$ in $\mathrm{cm}^{-3}$ and $S_{dry}$ in $\mathrm{m}^2\mathrm{cm}^{-3}$. For the values of the conversion parameters ($c_{250,d}$, $c_{s,d}$, $c_{250,c}$ and $c_{s,c}$) see Table 2.

| Parameterization | Eq. |
|---|---|
| Dust: | |
| $n_{250,d,dry} = c_{250,d}, \times \alpha_d$ | (8) |
| $S_{d,dry} = c_{s,d} \times \alpha_d$ | (9) |
| Non-dust, continental: | |
| $n_{250,c,dry} = c_{250,c} \times \alpha_c$ | (10) |
| $S_{c,dry} = c_{s,c} \times \alpha_c$ | (11) |

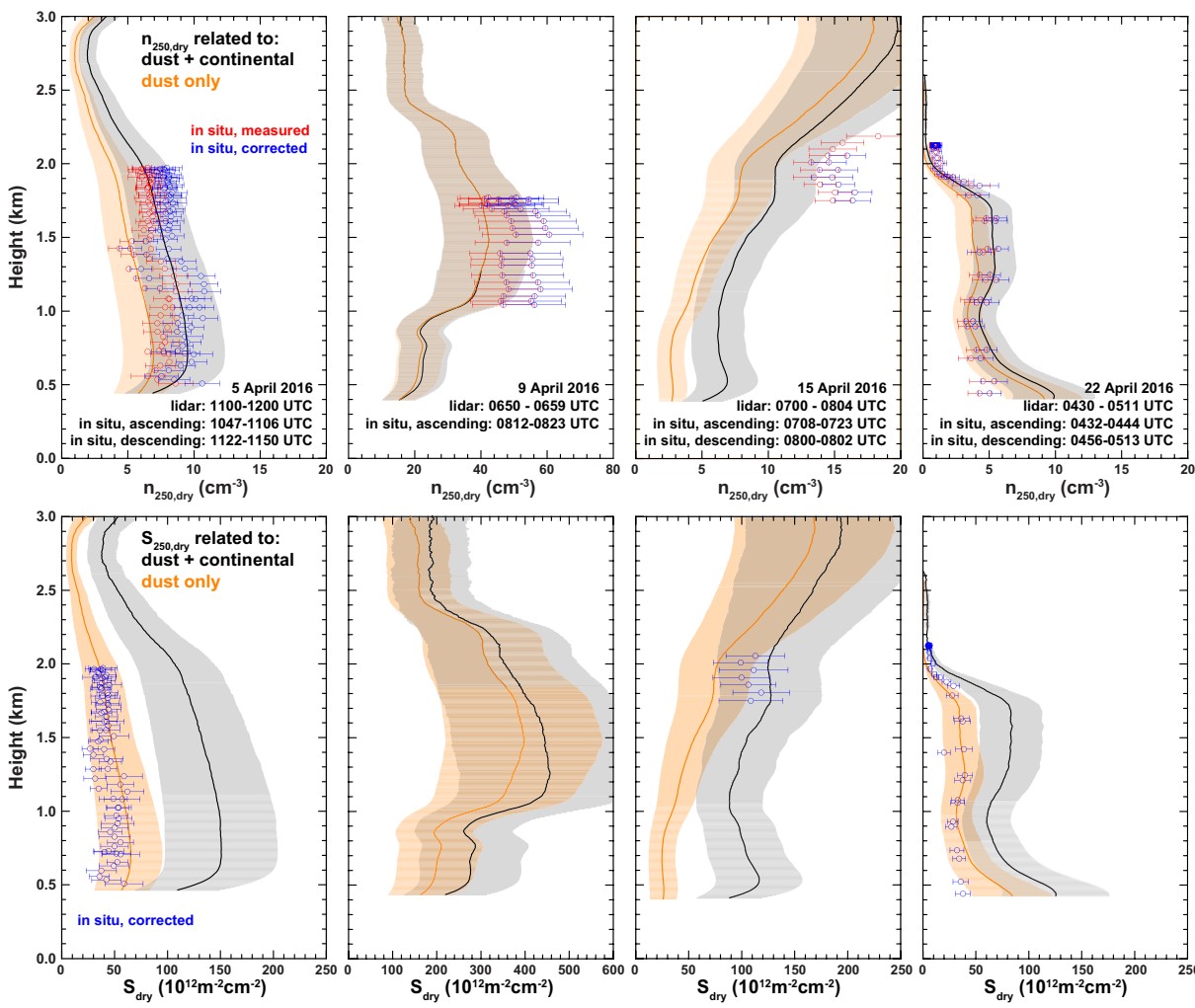

**Figure 7.** Profiles of $n_{250,dry}$ (upper panel) and $S_{dry}$ (lower panel) obtained from PollyXT and in-situ measurements (UAV uncorrected data in red, UAV corrected data in blue) on 5, 9, 15 and 22 April 2016. The lidar-derived profiles refer to dust only concentrations (orange), as well as the combination of dust and continental pollution concentrations (black).

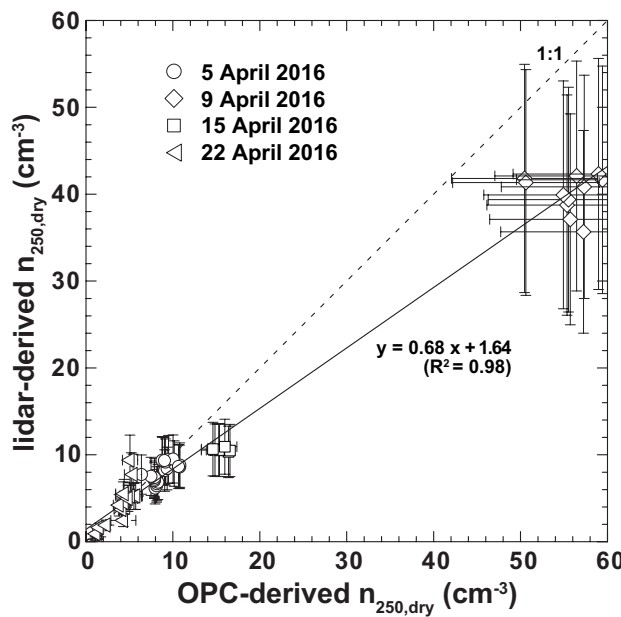

**Figure 8.** Correlation plot of $n_{250,dry}$ obtained from drone-based OPC measurements and inferred from lidar observations (values for a mixture of mineral dust and continental pollution, black in in Figure 7) during coordinated activities on 5, 9, 15 and 22 April 2016. The solid line marks the linear regression with the corresponding function and squared correlation coefficient given in the plot. The 1:1 line is given as dashed line.

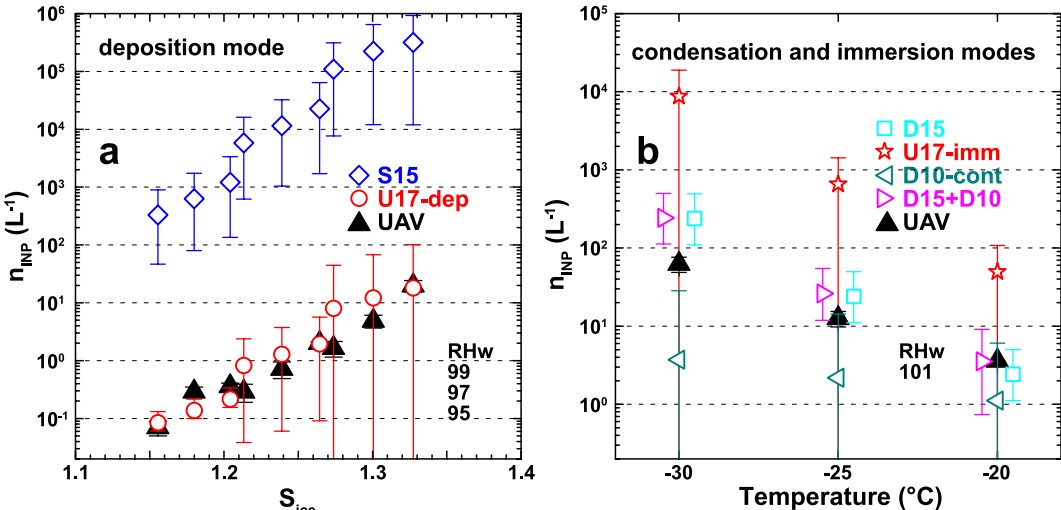

**Figure 9.** INP concentrations ($n_{INP}$) estimated from the CALIPSO lidar measurements on 21 April 2016 presented in Figure 6 (coloured symbols) and the UAV-FRIDGE measurements (black triangles) for (a) deposition freezing (as a function of saturation over ice) and (b) condensation and immersion freezing (as a function of temperature). Data in (a) are obtained for values of relative humidity over water of 95%, 97%, and 99%, leading to three values of $S_{ice}$ for each analysed temperature. A relative humidity over water of 101% is used to obtain the values presented in (b).

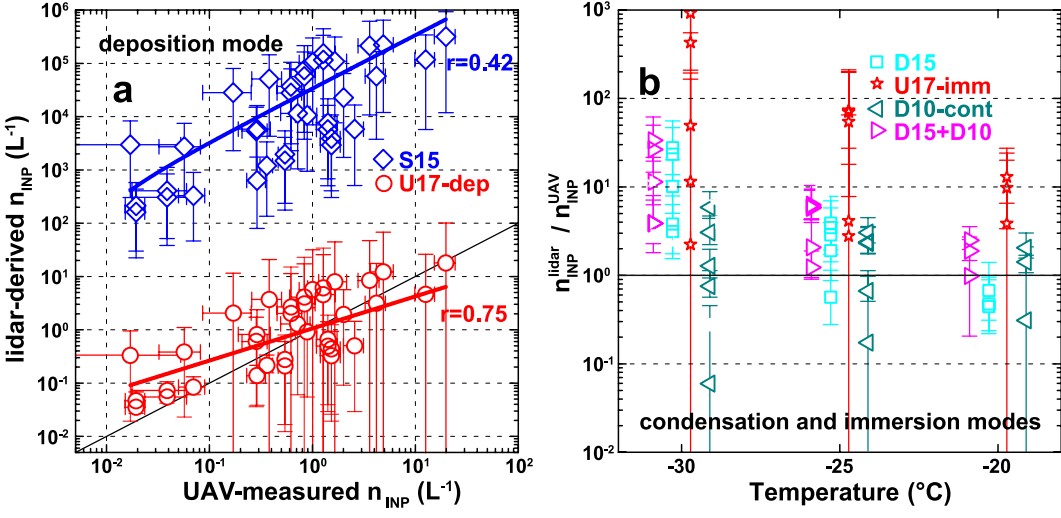

**Figure 10.** Comparison of INP concentrations derived from the CALIPSO and PollyXT lidar observations and UAV-FRIDGE measurements for (a) deposition freezing and (b) condensation and immersion freezing for cases with dust and continental presence. Colours and symbols refer to the used parameterization. Lines in (a) and (b) mark the 1:1 line. Numbers in (a) give Pearson's $r$ of the linear fits.

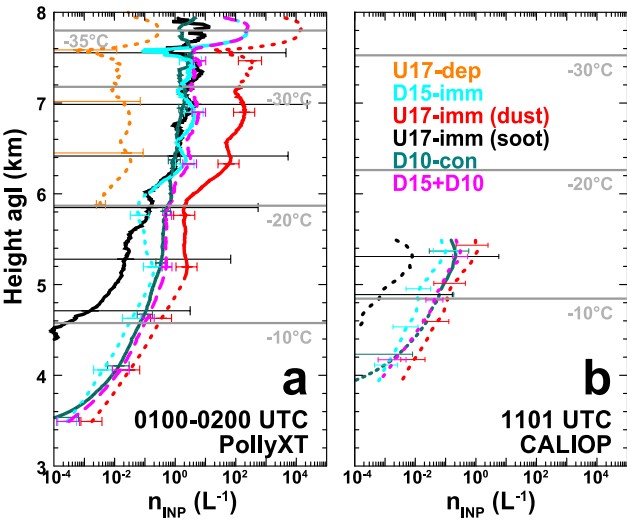

**Figure 11.** INP concentration profiles estimated from the measurements with (a) PollyXT between 01:00 and 02:00 UTC on 21 April 2016 and (b) CALIOP at 11:01 UTC on 21 April 2016. Temperature levels are derived from the WRF and MERRA-2 models. Colours refer to different INP parameterisations. Solid lines mark the temperature range for which the corresponding parameterisation has been developed. Dashed lines refer to the extrapolated temperature range (see Table 1).

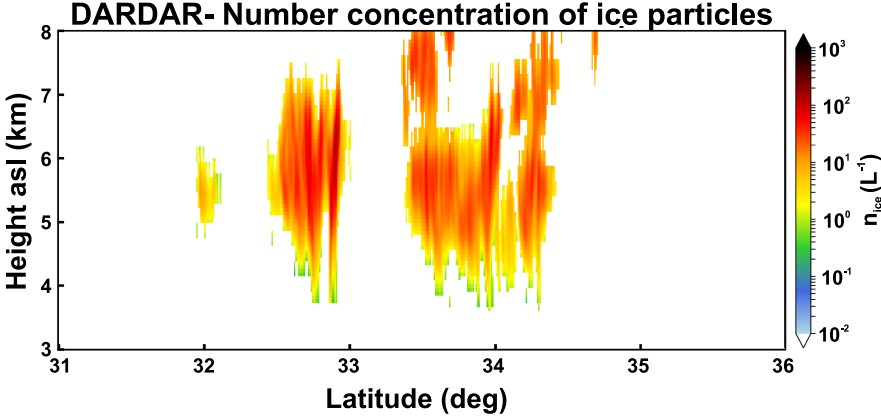

**Figure 12.** Spatial distribution of the DARDAR ice particle number concentrations at 11:01 UTC on 21 April 2016.

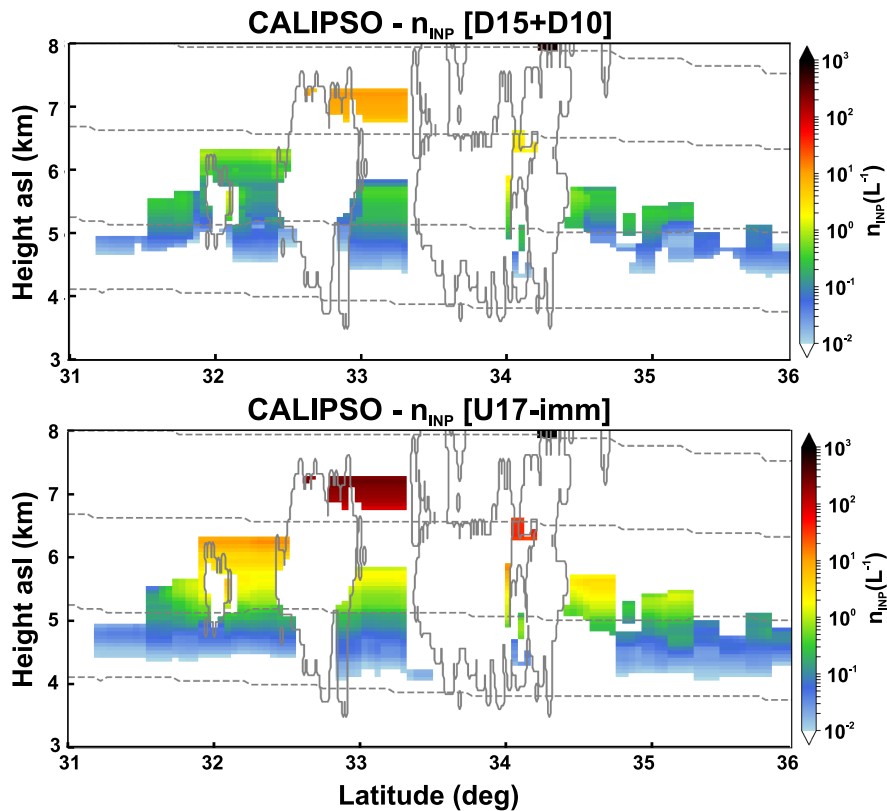

**Figure 13.** Spatial distribution of the INP concentrations during the event of 21 April 2016 at 11:01 UTC, as derived with the D15+D10 (top) and U17-imm (bottom) parameterisations. The location of the clouds observed are depicted with gray contours. The dotted lines correspond to T = 0, -10, -20 and -30 °C, based on the MERRA-2 model.

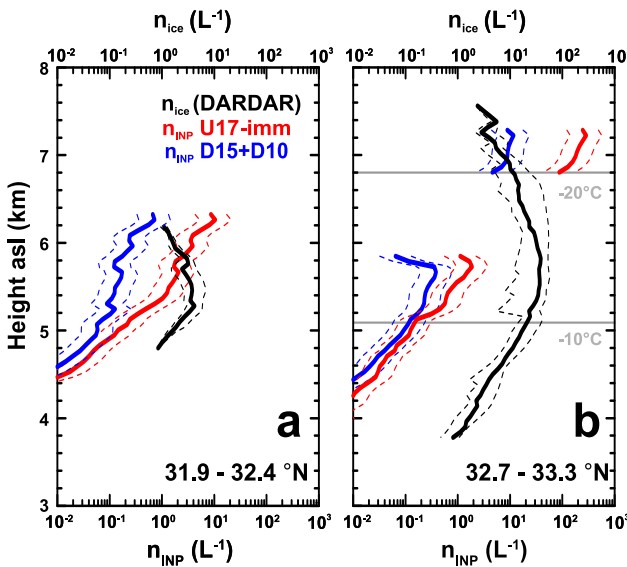

**Figure 14.** Concentration profiles of $n_{INP}$ and $n_{ice}$ from the A-Train measurements presented in Figure 12 and Figure 13 for the areas of (a) 31.9 to 32.4 °N (left) and (b) 32.7 to 33.3 °N (right). The $n_{INP}$ dotted lines denote the uncertainties of the estimations. The $n_{ice}$ dotted lines correspond to the 25 and 75% percentiles of the concentrations retrieved in the cloud. The overall uncertainty of the retrievals is discussed in the manuscript. The indicative temperature lines are from the MERRA-2 model.

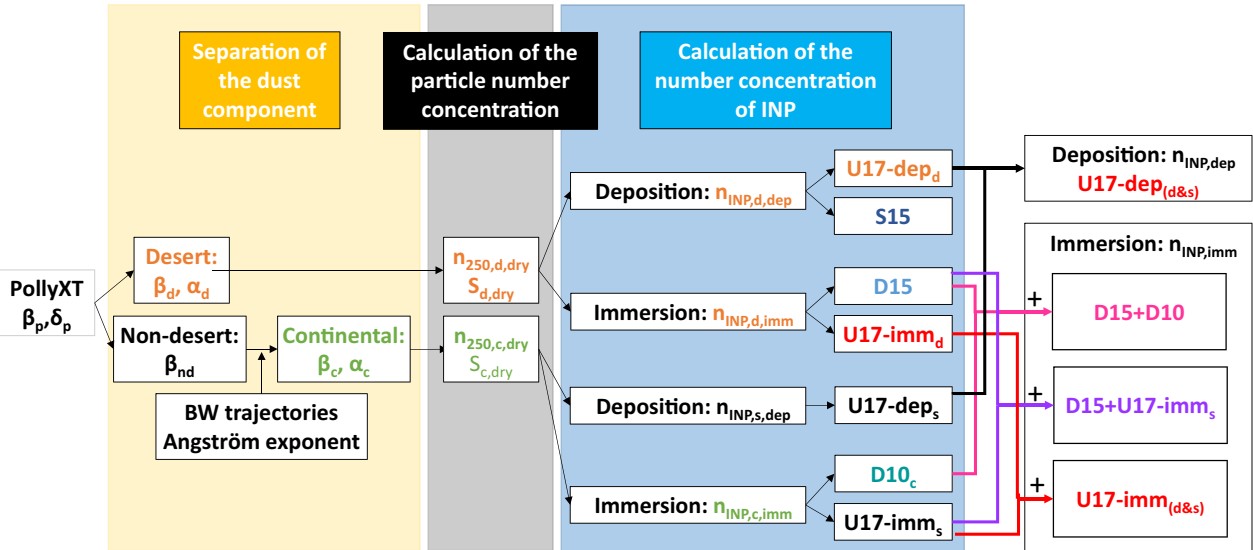

**Figure A1.** Overview of the data analysis scheme followed for the PollyXT measurements in this work. In the first step, we separate desert and non-desert backscatter coefficients ($\beta_d$ and $\beta_{nd}$) by means of the particle linear depolarization ratio ($\delta_p$). The backscatter coefficients for the non-desert aerosol is estimated to be continental aerosol mixtures $\beta_c$ by means of, e.g., backward (BW) trajectory analysis and Ångström exponent information. The two backscatter coefficients are then converted to aerosol-type-dependent particle extinction coefficients ($\alpha_i$). In the next step, the extinction coefficients are converted to aerosol-type-dependent profiles of particle number concentrations ($n_{250,i,dry}$) and particle surface area concentration ($S_{i,dry}$). In the next step, ice-nucleating particle number concentrations ($n_{INP,i}$) are estimated by applying INP parameterisations from the literature indicated by D10, D15, S15, U17 for DeMott et al. (2010), DeMott et al. (2015), Steinke et al. (2015) and Ullrich et al. (2017), respectively. Finally, the INP concentrations estimated for the different aerosol types are summed in order to estimate the total $n_{INP}$.

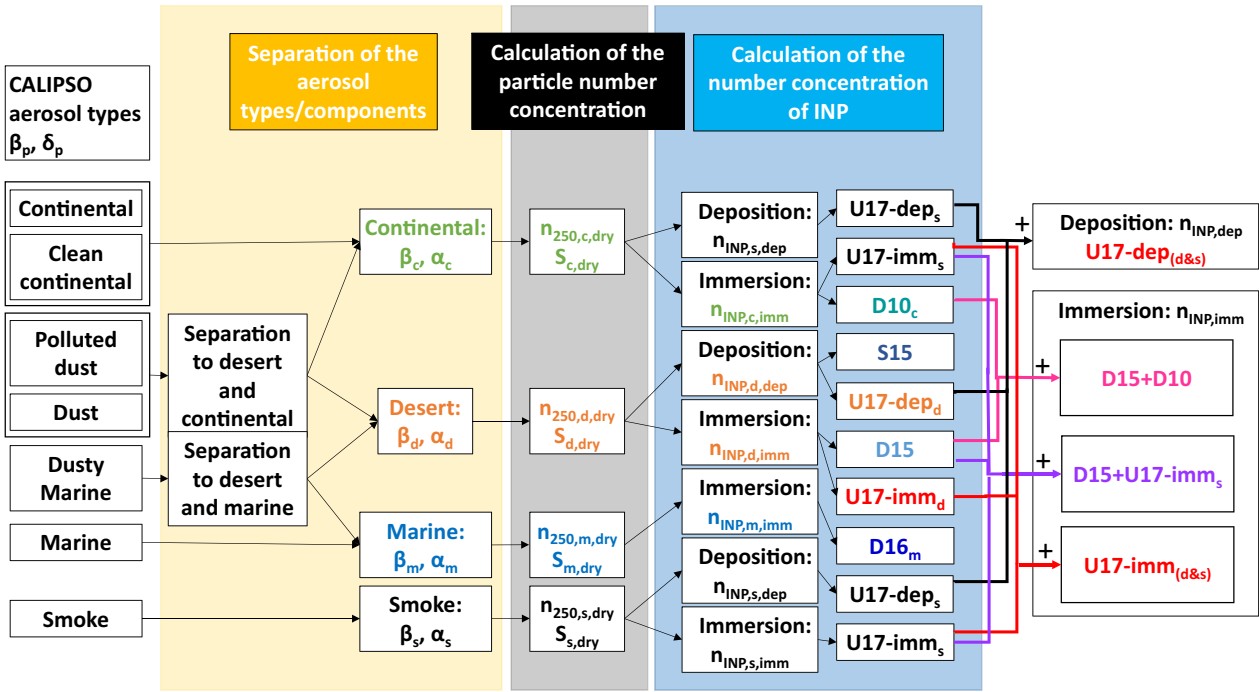

**Figure A2.** Overview of the data analysis scheme applied to CALIPSO measurements. In the CALIPSO case considered in this work only dust and polluted dust aerosol types have been observed. For that reason, only these combinations are considered here.