# Peer review of "Retrieval of ice nucleating particle concentrations from lidar observations and comparison with UAV in-situ measurements"

_Atmospheric Chemistry and Physics, 2018_

## Referee Comment (RC1) · Anonymous Referee #1 · 11 Jan 2019

General comments:

Marinou et al. show nicely the combination of field and laboratory work. Aerosol surface area concentrations derived from lidar observational data are used together with ice nucleation parameterizations derived from laboratory experiments to determine vertical profiles of aerosol-specific INP concentrations. The method is not know, but the authors included more state-of-the-art ice nucleation parameterizations and compared the INP number concentrations with offline analyzed filter samples taken with an UAV.

The comparison looks very promising. The authors show both immersion freezing and deposition nucleation nINP, although in the presented cases deposition nucleation

would be very unlikely. It would be nice to see a follow-up study for a real deposition nucleation case.

The manuscript is well structured, however some paragraphs are unnecessarily long, e.g. description of the differences of the parameterizations. Your focus is the case 20-22 April. So in my opinion you should shorten section 4.1 (description of the other cases) or discuss the other cases similarly.

The manuscript is well written, but I would propose to the authors going carefully through the paper and eliminate the typos and grammar error (some of them listed in the Technical comments section).

Specific comments:

1. Abstract A major point in your work is the comparison with the FRIDGE INP measurements from filters taken with a UAV. However, this is not mentioned in the abstract.

2. p. 2 l. 19 "about 1 in a million aerosol particles act as INP" This statement is well known, but I would prefer a reference.

3. p. 3 l. 1-3 This finding is not limited to field studies.

4. p. 4 l. 5 As far as I see, this listing is general. If so, than you might add the review by Murray et al. (2012) for another soot (immersion freezing) parameterization.

5. Table 1 First, to increase consistency you should use either K or degC. Second, the parameterization function U17-imm dust is wrong, if T is in K (as in the other equations)!

6. p. 4 l. 33-35 This statement is true, but D15 uses for its parameterization next to lab data also field data and therefor, the explanation for the discrepancy is not appropriate.

7. p. 5-6 First, the ordering is confusing, because the two nucleation mechanisms are mixed. Second, for the reader community a less technical description of the parameterizations would be valuable. It is obvious that soot and dust have a different ice nucleation behavior. I would suggest discussing the differences of the parameterizations in terms of the future outcome in your study. That means, when S15 shows a significant higher activated fraction then you would expect that the number of INP is much higher than for the U17-dep dust.
However, the error discussion is very good.

8. p. 7 l. 3 "(. . .) several in-situ instruments were operated" for what? What did they measure? Be more specific or remove that, because you do not use these instruments.

9. p. 8 l. 10 Can you give a reference for the SAMUM experiment?

10. p. 8 l. 11 This number was not given in percent, right?

11. Section 4 and 4.1 In the very first part of Section 4 you describe detailed the case of 21 April, but not the other cases. These cases are discussed in Section 4.1. This is confusing for the reader.

12. p. 11 l. 27-28 What was the height for 5 April?

13. p. 11 l. 34, Figure 7 Ok, but there is a deviation from the 1:1 line especially for the high concentrations (or case 9 April). Do you have an explanation or can you comment on that?

14. p. 13, Figure 8 and 9 The discussion of the two figures is quite similar and at some point you repeat the findings. Maybe you can shorten this part.

15. p. 13 l. 18ff You did not this detailed discussion for deposition nucleation. Be more consistent.

16. p. 15, Figure 10 From the campaign, are there temperature and/or relative humidity measurements available e.g. from radiosondes? From the WRF temperature profiles, you could argue that deposition nucleation will not be the case for your study. Furthermore, you could add the approximate cloud base and top height in Figure 10.

17. Summary section The conclusion are very short. Maybe you can discuss in more detail what improvements you or the community can do to improve the outcome, e.g. collocated temperature/ humidity profiling for calculating the INP concentration at real conditions, or combined in-situ ice concentration measurements.

Technical corrections:

1. p. 1, l. 6 Either "(. . .) lidar measurements with a INP efficiency (. . .)" or "(. . .) lidar measurements with INP efficiency parameterizations (. . .)"

2. p. 1 l. 12  14 agrees

3. p. 1 l. 12 nINP not yet introduced

4. p. 2 l. 6 "Our analysis" either has shown or shows "that (. . .)"

5. p. 2 l. 8 gives

6. p. 2 l. 8 Neither n250,dry nor Sdry introduce

7. p. 2 l. 30 citation style in the brackets

8. p. 3 l. 32 UAV comes first here, write out in full

9. p. 4 l. 3 citation style

10. p. 4 l. 16 AIDA comes first here, write out in full

11. p. 4 l. 27 "need to be transferred"

12. p. 4 l. 32 "(. . .) from Arizona, which have been (. . .) and are much more (. . .)"

13. p. 5 l. 5 devices

14. p. 5 l. 7 citation style

15. p. 5 l. 13 shown

16. p. 6 l. 4 desert

17. p. 6 l. 10, 13, 16 5 degC

18. p. 9 l. 28 "(. . .) and the Arabian Peninsula to the Eastern Mediterranean (. . .)"

19. p. 11 l. 28 seems

20. p. 12 l. 24 "(. . .) microscopy, which shows that (. . .)"

21. p. 13 l. 30 than instead of that

22. Figure 2 right figure Sdry has a wrong unit

---

## Referee Comment (RC2) · Anonymous Referee #2 · 14 Jan 2019

This study introduces a new methodology for detecting INP, which combines several information sources: the INP concentration profiles derived by lidar measurements; their comparison with UAVs measurements, and use of INP parameterizations for different freezing mechanisms. Necessary thermodynamic parameters are obtained from an atmospheric model.

The proposed approach contributes to better understanding of the complex process of cold cloud formation - one of the emerging issues attracting substantial attention of the scientific community. The article's subject is clearly presented with conclusions of high scientific relevance. It is well structured and provides detailed evidence on the IN

subject published in the community. However, in order to more improve the quality of the paper, I invite the authors to consider the following recommendations, comments and questions prior to the publication of the article:

Section 4 Results and discussion is the most important part of the paper which describes in details the evaluated results of the study. Figure 4 is excellent way to introduce a reader, in general, on considered processes (aerosol, clouds) over the selected observation period. After presenting evaluation of n250 and nINP retrievals (4.1 and 4.2 sub-sessions), Figure 10 is summarizing the major results of the proposed methodology. However, I find it not sufficient to promote the full value of the study. Namely, the figure shows INPs for only two instances of the 2-day selected period. From the figure, it is not possible to have evidence on the INP time evolution over the period, and also to conclude how INP correlates to cloud observations. I therefore ask the authors to generate a time-height INP graph (of a similar format as the one in Figure 4). For this additional result, PollyXT data should be adjusted to the time/height output of the WRF thermodynamic parameters. Consider also to compare the evaluated results with some satellite relevant cloud-related data such as e.g. ice water path, in order to show if the proposed method indicates the occurrence of cold clouds.

There are few minor issues to be also considered:

I suggest the following more concise article title: Retrieval of ice nucleating particle concentrations from lidar observations and comparison with UAVs measurements

Please provide more details on the WRF model data used to complement the observations: reference; resolution; source of the data; are the model temperature and humidity both used in the calculations?

P9 L22-23: 'Figure 4 provides an overview of the times and heights of the PollyXT and CALIPSO lidar measurements, along with the UAV measurements, between 20 and 22 April 2016'. What are the arguments that this particular period of observations is selected for detailed analysis but not some other similar ones during the April campaign?

What is the acronym OPC?

P10 L21: It should be useful to also show the volume depolarization ratio image.

Figure 4: Please include date markers on the x-axe (together with corresponding times)

Figure 1 (left) Why D15-dust is shown for T>-18C which is out of the validity range of this parameterization? Similar done for U17. Are the dashed lines extrapolations of D15 and U17? Please comment in the text.

Specify what are continental aerosols Figure 3: Include please a reference for the selected bimodal distribution, if any

---

## Author Comment (AC1) · 29 May 2019

Marinou et al. show nicely the combination of field and laboratory work. Aerosol surface area concentrations derived from lidar observational data are used together with ice nucleation parameterizations derived from laboratory experiments to determine vertical profiles of aerosol-specific INP concentrations. The method is not know, but the authors included more state-of-the-art ice nucleation parameterizations and compared the INP number concentrations with offline analyzed filter samples taken with an UAV.

The comparison looks very promising. The authors show both immersion freezing and deposition nucleation nINP, although in the presented cases deposition nucleation would be very unlikely. It would be nice to see a follow-up study for a real deposition nucleation case.

[REPLY] We thank the reviewer for his/her careful reading, comments and suggestions, which we address in the following. With his/her suggestions, we believe that the new version of the manuscript is significantly improved. The author's replies along with the changes in the manuscript are listed below.

> Remark: The figure numbers and the page numbers in the referee comments are corresponding to the original manuscript. If not stated otherwise, figure and page numbers in the authors' answers are referring to the revised, marked-up manuscript version (showing the changes made) which can be found attached to this answer.

**General comments:**

The manuscript is well structured, however some paragraphs are unnecessarily long, e.g. description of the differences of the parameterizations. Your focus is the case 20-22 April. So in my opinion you should shorten section 4.1 (description of the other cases) or discuss the other cases similarly.

[REPLY] We moved the description of the other cases from section 4.1 to the beginning of section 4 (so now all the cases are together) and we shorten their description. The new text is provided in the reply of comment nr.11 below. Additionally, we include in the text the arguments that characterize the event of 20-21 April as our golden case and the reason that we separate that case from the rest cases by adding:

In page 12 line 24 "The pure dust event on 20 to 21 April 2016 is considered the golden case of our dataset, as it has been observed simultaneously with the PollyXT lidar, the UAVs and the A-Train satellites. Additionally, it is the only pure-dust event of our dataset where we have simultaneously good lidar observations and in-situ INP measurements."

In page 16 line19 "The sample of 21 April … This sample is used in order to evaluate the performance of the $n_{INP}$ lidar estimates in a pure dust case, where (i) the errors originating from the first step of our methodology (separation in dust and non-dust aerosol components) are small ($\sim$ 30%) and (ii) the uncertainties induced from the D10 and U17-(soot) parameterizations are minimum."

The manuscript is well written, but I would propose to the authors going carefully through the paper and eliminate the typos and grammar error (some of them listed in the Technical comments section).

[REPLY] We read carefully the paper and eliminate the typos and grammar error that we find along with the ones that are listed in the technical comments section. Thank you very much for this.

**Specific comments:**

**1. Abstract A major point in your work is the comparison with the FRIDGE INP measurements from filters taken with a UAV. However, this is not mentioned in the abstract.**

[REPLY] We thank the reviewer for this comment. We add tis information in the abstract.

New version: page 1, line 7: "Here, we assess the feasibility of this new method for both ground-based and space-borne lidar measurements, using in-situ observations collected with Unmanned Aerial Vehicles (UAVs) and subsequently analyzed with the FRIDGE (FRankfurt Ice nucleation Deposition freezinG Experiment) INP counter from an experimental campaign at Cyprus in April 2016."

**2. p. 2 l. 19 "about 1 in a million aerosol particles act as INP" This statement is well known, but I would prefer a reference.**

[REPLY] We add the reference of Nenes et al. (2014): Nenes, A., Murray, B., and Bougiatioti, A.: Mineral Dust and Its Microphysical Interactions with Clouds, In Knippertz, P., and Stuut, J.B., Mineral Dust: A Key Player in the Earth System, pp. 287-325, Springer, ISBN 978-94-017-8977-6, 2014.

New version: page 2, line 24: ".. (about one particle in a million act as INP; Nenes et al. (2014)).."

**3. p. 3 l. 1-3 This finding is not limited to field studies.**

[REPLY] You are right. We corrected the phrasing of this sentence.

New version: page 3, line 8: "Observational studies have shown that immersion freezing dominates at temperatures higher than -30°C, while deposition nucleation dominates below -35°C (Ansmann et al., 2008, 2009; Westbrook et al., 2011; de Boer et al., 2011)."

**4. p. 4 l. 5 As far as I see, this listing is general. If so, than you might add the review by Murray et al. (2012) for another soot (immersion freezing) parameterization.**

[REPLY] We add the Murray et al. (2012) parameterization in the sentence.

**5. Table 1 First, to increase consistency you should use either K or degC. Second, the parameterization function U17-imm dust is wrong, if T is in K (as in the other equations)!**

[REPLY] We thank the reviewer for this comment. We corrected the U17-imm dust formula in the table. Now in all the equations T is in K.

**6. p. 4 l. 33-35 This statement is true, but D15 uses for its parameterization next to lab data also field data and therefor, the explanation for the discrepancy is not appropriate.**

[REPLY] I cannot find the explanation for discrepancy the reviewer refers to in the text in page 4 line 33-35 (or around). Please send me the specific extract from the manuscript. In

the meanwhile, in the new version we emphasized the use of filed data for the D15 parameterization and we rephrased the discussion on the S15 enhanced freezing efficiency, so we consider the sentences more clear and accurate.

New version: page 4, line 35: "Additionally, the parameterization of DeMott et al. (2015) (D15) (Table 1; Eq. 2) addresses the immersion and condensation freezing activity of natural mineral dust particles based on laboratory studies using the continuous flow diffusion chamber (CFDC) of the Colorado State University's and **field data from** atmospheric measurements in Saharan dust layers."

New version: page 5, line 11: "... S15 (Table 1; Eq. 4) was based on dust samples from Arizona, which were treated (washed, milled, treated with acid) and are much more ice active than **natural** desert dusts **particles** on average. Although S15 parameterization was based on **"treated"** dust samples which usually show an enhanced freezing efficiency, it is used in the NMME-DREAM model (Non-hydrostatic Mesoscale Model on E grid, Janjic et al. (2001); Dust REgional Atmospheric Model, Nickovic et al. (2001); Pérez et al. (2006)) for INP concentration estimations (Nickovic et al. , 2016). For this reason, it is included in this work."

**7. p. 5-6 First, the ordering is confusing, because the two nucleation mechanisms are mixed. Second, for the reader community a less technical description of the parameterizations would be valuable. It is obvious that soot and dust have a different ice nucleation behavior. I would suggest discussing the differences of the parameterizations in terms of the future outcome in your study. That means, when S15 shows a significant higher activated fraction then you would expect that the number of INP is much higher than for the U17-dep dust. However, the error discussion is very good.**

[REPLY] we rephrase the section according the instructions of the reviewer. The new section is:

New version: page 5, line 30: "Figure 1 provides an indication of the relative differences of the observed $n_{INP}$ in nature for immersion (right) and deposition (left) modes and in relation with the different aerosol compositions by showing a summary of the different nINP parameterizations. Specifically, the plot shows the fraction of the ice-activated particles ($f_i$ = $n_{INP}/n_{50,dry}$) for desert dust (dark blue, orange, red, light blue), continental (green) and soot (black). The particle concentrations used here, are derived assuming an extinction coefficient of 50 $Mm^{-1}$ for each of the different aerosol types (dust, continental, soot). The shaded areas take into account a range of the extinction coefficient from 10 $Mm^{-1}$ (lower limit) to 200 $Mm^{-1}$ (upper limit). The error bars mark the cumulative error in $f_i$ that results from the uncertainty in the lidar observations and their conversion to mass concentration as well as from the errors in the respective parameterizations. An overview of the typical values and the uncertainties used for the error estimation in this study is provided in Table 2. The deposition nucleation estimations in the left panel of Figure 1 are provided for $ss_i$ = 1.15 (solid lines) and $ss_i$ = (1.05, 1.1, 1.2, 1.3, 1.4) (dashed lines) to give a perspective on the range of possible values. Note here that although the immersion parameterizations were obtained using measurements at the temperature ranges of [-30, -14]°C (U17-imm, dust), [-35, -21]°C (D15, dust), [-34, -18]°C (U17-imm, soot) and [-35, -9]°C (D10, continental), they are extrapolated herein to extend over the immersion-freezing temperature range (dashed part of the lines in the immersion mode chart).

Figure 1 (left panel) shows that, for deposition mode, the dust ice-activated fractions from S15 are several orders of magnitude higher than those of U17-imm (e.g. 4 orders of magnitude at -40°C and $ss_i$ = 1.15%). Furthermore, the deposition 5ice-activation fraction of dust and soot (from U17-dep) differ significantly with soot being more active than dust for T <-38°C (up to 2 orders of magnitude) and dust being more active than soot for T >-38°C (up to 4 orders of magnitude).

Figure 1 (right panel) shows that, for immersion mode, the dust ice-activated fractions obtained from D15 are one order of magnitude lower than those calculated with U17-imm. Laboratory ice nucleation measurements and corresponding instrument inter-comparisons, have shown that at a single temperature between two and four orders of magnitude differences are observed as a result of the natural variability of the INP active fraction (DeMott et al., 2010, 2017) or the use of different INP counters (Burkert-Kohn et al., 2017). Hereon, we consider D15 and U17-imm as the lower and upper bounds of the immersed $n_{INP}$ estimations for dust INP populations. Figure 1 (immersion mode panel) illustrates the dust activation increase of up to six orders of magnitude within the mixed-phase temperature regime (−15 °C to −35 °C). For a 5 °C decrease, $n_{d,INP}$ increases by about one order of magnitude. Moreover, we see that at T < -18°C the immersion freezing desert dust ice activation (D15) is higher than the continental one (D10) while this changes at T > -18°C. On the contrary, soot (U17-imm) has always lower $f_i$ than dust (from either D15 or U17-imm). The ice-activated fractions of continental (D10) and soot (U17-imm) aerosols have a relative difference that is always less than 60% at T < -18°C. At higher temperatures they diverge with continental fi to exceed the soot one by one order of magnitude at T > -11∘C."

**8. p. 7 l. 3 "(…) several in-situ instruments were operated" for what? What did they measure? Be more specific or remove that, because you do not use these instruments.**

[REPLY] Thank you very much for this suggestion. We remove that part from the sentence, as indeed we do not use any of these data here.

New version: page 7, line 25: "An Aerosol Robotic Network (AERONET, Holben et al. 1998) sun photometer was located at the Cyprus Atmospheric Observatory of Agia Marina Xyliatou (35°02'19"N, 33°03'28"E, 532 m asl, 7 km west of the UAV airfield)."

**9. p. 8 l. 10 Can you give a reference for the SAMUM experiment?**

[REPLY] For a general reference of the SAMUM1 and 2 experiments, we add the reference: Ansmann, A., Petzold, A., Kandler, K., Tegen, I., Wendisch, M., Müller, D., Weinzierl, B., Müller, T., and Heintzenberg, J.: Saharan mineral dust experiments SAMUM-1 and SAMUM-2: what have we learned?, Tellus B, 63, 403–429, doi:10.1111/j.1600- 0889.2011.00555.x, 2011b.

New version: page 9, line 5: "… This assumption has been validated against airborne in-situ observations of the particle size distribution during the Saharan Mineral Dust Experiment (SAMUM; Ansmann et al. (2011b)) in Morocco."

**10. p. 8 l. 11 This number was not given in percent, right?**

[REPLY] Yes, you are right. We corrected it in the manuscript.

New version: page 9, line 7: "The correlation drops to ≈0.85±0.10 for urban environments based on ground-based in-situ measurements of particle size distributions at the urban site of Leipzig (Mamouri and Ansmann, 2016)."

**11. Section 4 and 4.1 In the very first part of Section 4 you describe detailed the case of 21 April, but not the other cases. These cases are discussed in Section 4.1. This is confusing for the reader.**

[REPLY] We thank the reviewer for his comment. He is right, so we move the discussion of all the cases in the beginning of Section 4 (from page 11 line 32 to page 13 line 12), and in the new version only the comparison between the UAV-measured and lidar-derived concentrations are discussed.

**12. p. 11 l. 27-28 What was the height for 5 April?**

[REPLY]  We add the following sentence in the manuscript:

New version: page 15, line 2: "These measurements correspond to heights above 0.5 km on 5th of April."

**13. p. 11 l. 34, Figure 7 Ok, but there is a deviation from the 1:1 line especially for the high concentrations (or case 9 April). Do you have an explanation or can you comment on that?**

[REPLY] We add the following explanation in the manuscript:

New version: page 15, line 6: "On 9 April we observed the highest differences between the lidar-derived and in-situ-measured $n_{250,dry}$, which may be attributed to the ~1 hr time difference between the in-situ sampling and the lidar retrieval (limitation due to mid-level clouds as discusses already). Nevertheless, the case is included here, as it represent the strongest dust event observed during the campaign."

**14. p. 13, Figure 8 and 9 The discussion of the two figures is quite similar and at some point you repeat the findings. Maybe you can shorten this part.**

[REPLY] In the new version, we have merged the discussion of these 2 figures (Figure 9 and 10 in the new version). The discussion is shorter in some extend and there are no repetitions that were present before.

New version:

page 17, line 9 – line 18: "Figure 9 (b) and Figure 10 (b) shows …"

page 17, line 19 – page 18, line 12: "In Figure 9 (b) and Figure 10 (b) we see that …"

**15. p. 13 l. 18ff You did not this detailed discussion for deposition nucleation. Be more consistent.**

[REPLY] We tried to be more consistent when discussing the immersion and deposition modes. In the new version, the deposition nucleation results are discussed in 10 lines (page 14, lines 19 – 28 in the final not marked up version) and the immersion/condensation results in 32 lines (page 14, line 29 – page 15, line 25 in the final not marked up version). The reason for the remaining difference is attributed to three things:

1. From the 2 existing deposition parameterization (S15, U17), we initially know that the S15 one is not good enough for natural desert dust (as it is based on treated dust samples with

modified ice activity) but we include it anyway for completeness purposes as it is currently used in the BSC-Dream model. On the other hand, the parameterization of U17-dep provided excellent agreement with the in-situ measurements; hence, we do need any discussion on disagreement with the in-situ (as there is none).

2. For immersion mode, there are many parameterization in the literature available (D15, D10, U17) which are based on natural aerosol measurements, but they provide different INPC results. Additionally, the differences observed, even for the same parameterization, varied a lot (from identical up to 3 orders of magnitude different than the in-situ INPC - when the samples are analyzed in different temperature). In the manuscript, we discuss these differences and the possible sources of discrepancies and errors of the in-situ measurements (FRIDGE is widely used for immersion measurements but it was originally constructed for deposition nucleation measurements and hence the deposition IN measurements are more accurate).

3. In the discussion of immersion/condensation INP estimates, we provide indicatively some INPC values of the in-situ measurements and the lidar retrievals for the case of 21 April, as is later on discussed in detail and in comparison with cloud $n_{ICE}$ observations in section 4.3. The relevant temperatures in this case are <-35°C hence only the immersion INP estimates are interesting to be mentioned.

**16. p. 15, Figure 10 From the campaign, are there temperature and/or relative humidity measurements available e.g. from radiosondes? From the WRF temperature profiles, you could argue that deposition nucleation will not be the case for your study. Furthermore, you could add the approximate cloud base and top height in Figure 10.**

[REPLY] The WRF modeled profiles used are assimilated with the NCEP global reanalysis dataset. We have included this information in section 3.3 (page 9, lines 18 – 25).

Also, we added the following argumentation in the discussion of this case (page 19, line 20): "From the WRF and MERRA-2 assimilations we see that T < -35°C in heights up to 7.8 km agl, which indicate that the immersion freezing mechanism is dominant in this case and that the deposition nucleation mechanism is not significant."

Furthermore, we add a new figure where we are indicating the cloud boundaries in this event next to the $n_{INP}$ values: new Figure 13.

**17. Summary section The conclusion are very short. Maybe you can discuss in more detail what improvements you or the community can do to improve the outcome, e.g. collocated temperature/ humidity profiling for calculating the INP concentration at real conditions, or combined in-situ ice concentration measurements.**

[REPLY] We included the following discussion in the conclusions:

New version: page 21, line 32:" A further step for improving the lidar-derived INP retrievals and investigating the different parametrizations used is by conducting dedicated studies with collocated lidar measurements and additional temperature and humidity profiling in order to calculate the INP concentrations at real conditions, and the combination of the retrieved n_INP with airborne in-situ ice concentration measurements."

**Technical corrections:**

1.  **p. 1, l. 6 Either "(…) lidar measurements with a INP efficiency (…)" or "(…) lidar measurements with INP efficiency parameterizations (…)"**

[REPLY] Corrected as "(…) lidar measurements with INP efficiency parameterizations (…)"

**2. p. 1 l. 12 14 agrees**

[REPLY] Corrected

**3. p. 1 l. 12 nINP not yet introduced**

[REPLY] Changed to "INP concentrations ($n_{INP}$)"

**4. p. 2 l. 6 "Our analysis" either has shown or shows "that (…)"**

[REPLY] Corrected to "shows"

**5. p. 2 l. 8 gives**

[REPLY] Corrected

**6. p. 2 l. 8 Neither n250,dry nor Sdry introduce**

[REPLY] The n250 was introduced in the previous paragraph and the Sdy in this line. But indeed, they were not comfortably understood while reading this part. We change this sentence and include their definitions in parentheses next to the symbols.

New version: page 3, line 22: "Lidar measurements can provide profiles of $n_{250,dry}$ (the number of aerosol particles with dry radius greater than 250nm) and $S_{dry}$ (the aerosol particles dry surface area concentration) related to mineral dust, continental pollution and marine aerosol..".

**7. p. 2 l. 30 citation style in the brackets**

[REPLY] Corrected

**8. p. 3 l. 32 UAV comes first here, write out in full**

[REPLY] done

**9. p. 4 l. 3 citation style**

[REPLY] Corrected

**10. p. 4 l. 16 AIDA comes first here, write out in full**

[REPLY] Corrected to "Aerosol Interaction and Dynamics in the Atmosphere (AIDA) cloud chamber"

**11. p. 4 l. 27 "need to be transferred"**

[REPLY] Corrected

**12. p. 4 l. 32 "(…) from Arizona, which have been (…) and are much more (…)"**

[REPLY] Corrected

**13. p. 5 l. 5 devices**

[REPLY] Corrected

**14. p. 5 l. 7 citation style**

[REPLY] Corrected

**15. p. 5 l. 13 shown**

[REPLY] Corrected

**16. p. 6 l. 4 desert**

[REPLY] Corrected

**17. p. 6 l. 10, 13, 16 5 degC**

[REPLY] Corrected

**18. p. 9 l. 28 "(…) and the Arabian Peninsula to the Eastern Mediterranean (…)"**

[REPLY] Corrected

**19. p. 11 l. 28 seems**

[REPLY] Corrected

**20. p. 12 l. 24 "(…) microscopy, which shows that (…)"**

[REPLY] Corrected

**21. p. 13 l. 35 than instead of that**

[REPLY] Corrected

**22. Figure 2 right figure Sdry has a wrong unit**

[REPLY] Thank you very much. We corrected the unit.

---

## Author Comment (AC2) · 29 May 2019

**This study introduces a new methodology for detecting INP, which combines several information sources: the INP concentration profiles derived by lidar measurements; their comparison with UAVs measurements, and use of INP parameterizations for different freezing mechanisms. Necessary thermodynamic parameters are obtained from an atmospheric model.**

**The proposed approach contributes to better understanding of the complex process of cold cloud formation - one of the emerging issues attracting substantial attention of the scientific community. The article's subject is clearly presented with conclusions of high scientific relevance. It is well structured and provides detailed evidence on the IN subject published in the community. However, in order to more improve the quality of the paper, I invite the authors to consider the following recommendations, comments and questions prior to the publication of the article:**

[REPLY] We thank very much referee #2 for his/her careful reading, comments and suggestions, which we address in the following. With his/her suggestions, we believe that the new version of the manuscript is significantly improved, and our findings are promoted in a better way. The author's answers along with the changes in the manuscript are listed below.

> Remark: The figure numbers and the page numbers in the referee comments are corresponding to the original manuscript. If not stated otherwise, figure and page numbers in the authors' answers are referring to the revised, marked-up manuscript version (showing the changes made) which can be found attached to this answer.

**General comments:**

**Section 4 Results and discussion is the most important part of the paper which describes in details the evaluated results of the study. Figure 4 is excellent way to introduce a reader, in general, on considered processes (aerosol, clouds) over the selected observation period. After presenting evaluation of n250 and nINP retrievals (4.1 and 4.2 sub-sessions), Figure 10 is summarizing the major results of the proposed methodology. However, I find it not sufficient to promote the full value of the study. Namely, the figure shows INPs for only two instances of the 2-day selected period. From the figure, it is not possible to have evidence on the INP time evolution over the period, and also to conclude how INP correlates to cloud observations. I therefore ask the authors to generate a time-height INP graph (of a similar format as the one in Figure 4). For this additional result, PollyXT data should be adjusted to the time/height output of the WRF thermodynamic parameters. Consider also to compare the evaluated results with some satellite relevant cloud-related data such as e.g. ice water path, in order to show if the proposed method indicates the occurrence of cold clouds.**

[REPLY] We thank the reviewer for this very constructive comment. We worked on these suggestions. We generated a new time height INP graph from the CALIPSO track (Figure 13) and we used the A-train observations during the event of 21-April-2016 in order to present the clouds observed above the scene and to evaluate the proposed methodology in the presence of ice clouds (new Figure 5, Figure 12 and Figure 14).

New version:

Page 10, line 30 – page 11, line 20: Section 3.4 "Space-borne cloud observations" describes the relevant space-borne data used to indicate the occurrence and ice number concentration of clouds.

Page 12, line 34 – page 13, line 6: "Figure 5 provides an overview of the aerosol and clouds above the area, with the MODIS true color image (upper panel) and the combined DARDAR and CALIPSO L2 feature mask (lower panel). Dust is observed above the broader region in altitudes up to 6 km and ice clouds are formed inside the dust layer South of Cyprus in altitudes greater than 4 km (T < 0◦C). The ice clouds are detected/characterized at 1 km horizontal resolution (DARDAR-MASK product) while the dust plume is detected at 20 and 80 km horizontal resolution (CALIPSO L2 product)."

Page 20, line 22 – Page 21, line 13: Presents the satellite relevant cloud-related estimations of ice number concentrations inside the clouds and the INP concentrations in their vicinity.

**There are few minor issues to be also considered:**

**I suggest the following more concise article title: Retrieval of ice nucleating particle concentrations from lidar observations and comparison with UAVs measurements**

[REPLY] We thank the reviewer for the suggestion. We change the title in: "Retrieval of ice nucleating particle concentrations from lidar observations and comparison with UAV in-situ measurements".

**Please provide more details on the WRF model data used to complement the observations: reference; resolution; source of the data; are the model temperature and humidity both used in the calculations?**

[REPLY] We thank the reviewer for this comment. We added the necessary information on the modeled fields used in the end of section 3.2. The paragraph added is the following:

New version: page 9, line 17: "In the final step, the $n_{INP}$ profiles are estimated using the ice nuclei parameterizations presented in Section 2 (Eq. (1)-(7)). For these calculations we are using collocated modeled profiles of the pressure, temperature and humidity fields. Specifically, for the PollyXT-based nINP calculations we use hourly outputs from the Weather Research and Forecasting atmospheric model (WRF; Skamarock et al. (2008)) which is operational at the National Observatory of Athens at a mesoscale resolution of 12 x 12 km and 31 vertical levels (Solomos et al., 2015, 2018). Initial and boundary conditions for the atmospheric fields and the 30 sea surface temperature are taken from the National Center for Environmental Prediction (NCEP) global reanalysis at 1◦x1◦ resolution. For the CALIPSO-bases nINP calculations we use the track-collocated meteorological profiles from the MERRA-2 model (Modern-Era Retrospective analysis for Research and Applications, Version 2) which are included in the CALIPSO V4 product (Kar et al., 2018)."

**P9 L22-23: 'Figure 4 provides an overview of the times and heights of the PollyXT and CALIPSO lidar measurements, along with the UAV measurements, between 20 and 22 April 2016'. What are the arguments that this particular period of observations is selected for detailed analysis but not some other similar ones during the April campaign?**

[REPLY] Thank you very much for this comment, indeed this information was missing. We added this information in the manuscript in:

New version: page 12 line 24: "The pure dust event on 20 to 21 April 2016 is considered the golden case of our dataset, as it has been observed simultaneously with the PollyXT lidar, the UAVs and the A-Train satellites. Additionally, it is the only pure-dust event of our dataset where we have simultaneously good lidar observations and in-situ INP measurements."

**What is the acronym OPC?**

[REPLY] OPC is "Optical Particle Counter". We made the first letters in each word capital in the manuscript.

New version: page 10, line 15: "Cruiser was additionally equipped with an Optical Particle Counter (OPC, Met One Instruments, Model 212 Profiler)."

**P10 L21: It should be useful to also show the volume depolarization ratio image.**

[REPLY] We updated Figure 4. The new figure has three images: the backscatter coefficient at 1064 nm, the volume depolarization ratio at 532 nm and the feature mask of the scene.

**Figure 4: Please include date markers on the x-axe (together with corresponding times)**

[REPLY] We included the date also in the markers.

**Figure 1 (left) Why D15-dust is shown for T>-18C which is out of the validity range of this parameterization? Similar done for U17. Are the dashed lines extrapolations of D15 and U17? Please comment in the text.**

[REPLY] Indeed the dashed lines are extrapolations of the parameterizations in the immersion-freezing range. This information is mentioned in the manuscript, but it is now additionally mentioned in the text of Figure 1.

New version: Figure 1 legend: "In the immersion mode (right) panel, the parameterizations are extrapolated over the immersion-freezing temperature range (dashed lines)".

**Specify what are continental aerosols Figure 3: Include please a reference for the selected bimodal distribution, if any**

[REPLY] We consider this sentence as an accidental merge of two separate commends: "**Specify what are continental aerosols**" and **"Figure 3: Include please a reference for the selected bimodal distribution, if any"**.

For the first one: "**Specify what are continental aerosols**"

[REPLY] We specify what is considered as continental aerosols in the following sentence:

New version: page 5, line 24: "As the majority of the samples used for D10 were non-desert continental aerosols, this INP parameterization has been considered to be suitable for addressing the immersion and condensation freezing activity of **mixtures of anthropogenic haze, biomass burning smoke, biological particles, soil and road dust** (Mamouri and Ansmann, 2016). **From here on these mixtures are addressed as continental aerosols.**"

For the second one: **"Figure 3: Include please a reference for the selected bimodal distribution, if any"**

[REPLY] Considering the choice of the bimodal size distribution (instead of e.g. a size distribution with more modes), we refer here to the work of Remer and Kaufinan (1998),

stating that physical processes in the atmosphere most frequently result in a bimodal structure of the aerosol size distribution. The specific bimodal size distribution is calculated as the best fit of the in situ measurements.

References:

Skamarock, W.C., Klemp, J.B., Dudhia, J., Gill, D.O., Barker, D.M., Duda, M.G., Huang, X.-Y., Wang, W., Powers, J.G.: A Description of the Advanced Research WRF Version 3. In: NCAR Technical Note 475, 2008, http://www.mmm.ucar.edu/wrf/users/docs/arw_v3.pdf, 2008.

Solomos, S., Amiridis, V., Zanis, P., Gerasopoulos, E., Sofiou, F.I., Herekakis, T., Brioude, J., Stohl, A., Kahn, R.A., Kontoes, C.: Smoke dispersion modeling over complex terrain using high resolution meteorological data and satellite observations – The FireHub platform, Atmospheric Environment, Volume 119, Pages 348–361, doi:10.1016/j.atmosenv.2015.08.066, 2015.

Solomos, S., Kalivitis, N., Mihalopoulos, N., Amiridis, V., Kouvarakis, G., Gkikas, A., Binietoglou, I., Tsekeri, A., Kazadzis, S., Kottas, M., Pradhan, Y., Proestakis, E., Nastos, P.T., Marenco, F. From Tropospheric Folding to Khamsin and Foehn Winds: How Atmospheric Dynamics Advanced a Record-Breaking Dust Episode in Crete, Atmosphere 2018, 9(7), 240, https://doi.org/10.3390/atmos9070240, 2018.

Kar, J., Vaughan, M. A., Lee, K. P., Tackett, J. L., Avery, M. A., Garnier, A., Getzewich, B. J., Hunt, W. H., Josset, D., Liu, Z., Lucker, P. L., Magill, B., Omar, A. H., Pelon, J., Rogers, R. R., Toth, T. D., Trepte, C. R., Vernier, J. P., Winker, D. M., and Young, S. A.: CALIPSO lidar calibration at 532 nm: version 4 nighttime algorithm, Atmos. Meas. Tech., 11, 1459-1479, https://doi.org/10.5194/amt-11-1459-2018, 2018.

---

## Author Comment (AC3) · 29 May 2019

The comment was uploaded in the form of a supplement:
https://www.atmos-chem-phys-discuss.net/acp-2018-1203/acp-2018-1203-AC3-supplement.pdf

---

## Author Response (AR2)

**accepted as is**

[REPLY] We thank very much referee #1 for his review.

**Anonymous Referee #2**

**The manuscript is well written, well structured and the subject is clearly presented with conclusions scientifically relevant to the scope of the journal.**

**The authors fulfilled successfully the comments and suggestions of the interactive discussion rewires. More specific, the revised manuscript includes the analysis of a case using combined aerosol and clouds satellite observations for the evaluation of the proposed methodology in the presence of ice clouds, providing the potential of the production of 3D products.**

**For the final publication, the authors could take into consideration the following minor remarks**

[REPLY] We thank very much referee #2 for his/her careful reading, comments and suggestions, which helped correct some unclear points and typos. The author's answers along with the changes in the manuscript are listed below.

> Remark: The figure numbers and the page numbers in the referee comments are corresponding to the original manuscript. If not stated otherwise, figure and page numbers in the authors' answers are referring to the revised, marked-up manuscript version (showing the changes made) which can be found attached to this answer.

1. **P11, L6-7: it is not clear how the constant $\delta_p$ during specific time period is evident of homogeneity between the two sites.**

[REPLY] We thank the reviewer for the suggestion. We refer to evidence of moderate variability in time. For that reason we clarify the sentence as:

New version: P11, L5-6: "In the next hours (until 12 UTC), only moderate variability was observed above the station (in the lidar backscatter coefficient and $\delta_p$ curtains - not shown)."

2. **P12, L23-24: Both Section 4.2 and Figure 9 present GB lidar-derived and UAV measurements comparison. I couldn't find any use of CALIPSO observations.**

[REPLY] We thank the reviewer for this comment. We have mentioned earlier (P12, L23-24) that: "The comparison between the CALIPSO-derived $n_{INP}$ and the UAV measurements from this case are discussed in Section 4.2 (see Figure 9)". We see that the readers could be confused if we do not mention it in Section 4.2 as well, hence we added this information in Section 4.2 and in the captions of Figures 9 and 10:

New version: Section 4.2 P14, L12-18: "Figure 9 shows the $n_{INP}$ on 21 April as they were calculated from the CALIPSO lidar measurements (colored symbols) and measured from the UAV-FRIDGE samples (black triangles), (a) for deposition nucleation (as a function of

saturation over ice) and (b) for condensation and immersion freezing (as a function of temperature).

Likewise, we are using all the aforementioned cases, in order to evaluate the performance of the $n_{INP}$ lidar estimates in cases with dust and continental aerosols. Figure 10 shows scatter plots of all the lidar-estimated $n_{INP}$ (from PollyXT and CALIPSO) against the in-situ measurements for (a) deposition nucleation and (b) condensation and immersion freezing."

Figure 9, caption: "INP concentrations ($n_{INP}$) estimated from the CALIPSO lidar measurements on 21 April 2016 presented in Figure 6 (coloured symbols) and the UAV-FRIDGE measurements (black triangles) …"

Figure 10, caption: "Comparison of INP concentrations derived from the CALIPSO and PollyXT lidar observations and UAV-FRIDGE measurements …"

3. **P17, L20: The authors may provide here the recent under discussion study of Ansmann et al 2019 as a reference.**

[REPLY] We included the under discussion study of Ansmann et al 2019.

New version: P17, L21: "However, further measurements are necessary to reach a more concrete conclusion, as for example, measurements of the atmosphere dynamics (e.g. from a wind lidar) and observations of the cloud evolution (e.g. from a cloud radar as in the recent study of Ansmann et al. (2019).

4. **Figure 9, caption line 1: the 21st of April 2016 profiles are not presented in Figure 7.**

[REPLY] We thank the reviewer for this comment. We corrected this typo. The profiles are presented in Figure 6.

[revised manuscript text omitted]